# Slow-Fast Policy Optimization: Reposition-Before-Update for LLM Reasoning

**Ziyan Wang**[1,†]   **Zheng Wang**[2,†]   **Xingwei Qu**[3]   **Qi Cheng**[4]   **Jie Fu**[5]   **Shengpu Tang**[6]
**Minjia Zhang**[2]   **Xiaoming Huo**[1]
[1] Georgia Institute of Technology    [2] University of Illinois Urbana-Champaign
[3] University of Manchester    [4] NVIDIA    [5] Independent    [6] Emory University
{wzy, huo}@gatech.edu   {zhengw10, minjiaz}@uiuc.edu

[†] Equal contribution

## Abstract

Reinforcement learning (RL) has become central to enhancing reasoning in large language models (LLMs). Yet on-policy algorithms such as Group Relative Policy Optimization (GRPO) often suffer in early training: noisy gradients from low-quality rollouts lead to unstable updates and inefficient exploration. We introduce Slow-Fast Policy Optimization (SFPO), a simple yet efficient framework to address the above limitations via decomposing each step into three stages: a short fast trajectory of inner steps on the same batch, a reposition mechanism to control off-policy drift, and a final slow correction. This reposition-before-update design preserves the objective and rollout process unchanged, making SFPO plug-compatible with existing policy-gradient pipelines. Extensive experiments demonstrate that SFPO consistently improves stability, reduces number of rollouts, and accelerates convergence of reasoning RL training. Specifically, it outperforms GRPO by up to 2.80 points in average on math reasoning benchmarks. It also achieves up to $4.93\times$ fewer rollouts and an up to $4.19\times$ reduction in wall-clock time to match GRPO's best accuracy. [1]

## 1 Introduction

Large language models (LLMs) have recently achieved remarkable progress on complex multi-step reasoning tasks, especially in mathematics problem solving and scientific question answering (OpenAI et al., 2024; DeepSeek-AI et al., 2025b). A central driver of these advances has been reinforcement learning (RL), which fine-tunes LLMs using reward signals tied to semantic correctness and solution quality. For example, DeepSeekMath (Shao et al., 2024) demonstrates that Group Relative Policy Optimization (GRPO) can substantially improve open-source LMs on mathematical reasoning, while DeepSeek-R1 (Guo et al., 2025) shows that RL signals alone can induce emergent reasoning behaviors across various domains.

Despite these successes, GRPO inherits structural inefficiencies that make training fragile in LLM reasoning. During the early stages of training, when rollouts are particularly weak or uninformative, stochastic rewards induce high-variance gradients that destabilize updates (Yu et al., 2025a; Zheng et al., 2025; Dai et al., 2025; Shen et al., 2025). Compounding this, common practice in GRPO implementations is to use a single optimization step per rollout batch, and such a one-shot update tends to produce a noisier, less reliable step update direction, especially at the early training stages, thereby underutilizing each batch and discarding information that could be further exploited (Schulman et al., 2017; Engstrom et al., 2020). Although the original GRPO paper (Shao et al., 2024) explored reusing rollout data, our findings suggest that simply applying this off-policy update to the policy model can degrade performance as training proceeds, and how to use it effectively to build strong reasoning abilities in LLMs with satisfying sample efficiency remains an open question.

---

[1] Project website is available at https://slow-fast-po.github.io/.

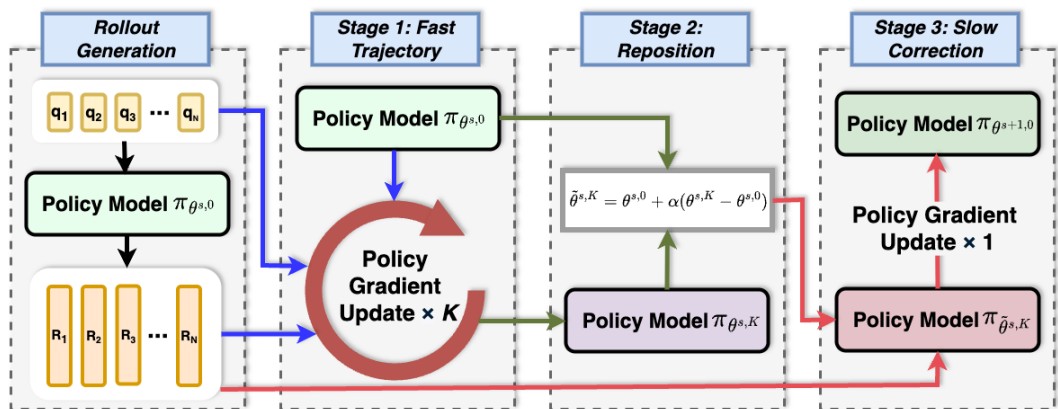

Figure 1: Pipeline of SFPO at iteration $s$. Starting from the current policy $\pi_{\theta^{s,0}}$, we first generate rollouts for training. **Stage I (Fast Trajectory):** apply $K$ successive gradient updates on the same batch to obtain $\theta^{s,K}$. **Stage II (Reposition):** interpolate between $\theta^{s,K}$ and the starting point $\theta^{s,0}$ to form $\widetilde{\theta}^{s,K}$, controlling off-policy drift. **Stage III (Slow Correction):** perform one additional update on $\widetilde{\theta}^{s,K}$, yielding $\pi_{\theta^{s+1,0}}$ for the next iteration.

In this work, we introduce **Slow–Fast Policy Optimization (SFPO)**, a simple yet effective enhancement to on-policy policy-gradient methods (e.g., GRPO) that stabilizes noisy updates and improves sample efficiency while leaving the base loss, rollout collection, and KL/clip regularization unchanged. Specifically, each training step is restructured into three coordinated stages: **(i) fast**—multiple inner updates on the same batch to stabilize the search direction; **(ii) reposition**—interpolation back toward the starting point to control off-policy drift; and **(iii) slow**—an extra gradient correction to align with local curvature. This reposition-before-update design transforms noisy one-shot updates into structured trajectories, yielding more stable optimization, higher sample efficiency, and faster convergence. We empirically validate the efficacy of SFPO through extensive experiments across five models and six reasoning benchmarks. The evaluation shows that SFPO outperforms GRPO by up to **2.80** points in average on math reasoning benchmarks. It also uses up to **4.93**× fewer rollouts while reducing wall-clock time by up to **4.19**× to reach GRPO's best accuracy. We summarize our contributions as follows:

- We propose SFPO, a plug-compatible update rule that reuses rollout batches via a fast–reposition–slow decomposition, with theoretical insights for each stage.

- SFPO introduces no changes to the underlying objective, rollout generation, or regularization, enabling drop-in integration into existing LLM reasoning pipelines.

- Through extensive experiments on math reasoning benchmarks, we show that SFPO consistently improves stability, sample efficiency, and convergence speed over GRPO.

## 2 PRELIMINARIES

**Group Relative Policy Optimization (GRPO)** (Shao et al., 2024) is a policy gradient method that has been widely applied in large-scale LLM training pipelines. GRPO dispenses with an explicit value function and instead normalizes rewards across a group of responses to the same prompt. Given an input $q$, the policy $\pi_\theta$ generates $G$ candidate sequences $\{o_i\}_{i=1}^{G}$ with corresponding rewards $\{r_i\}_{i=1}^{G}$. The normalized advantage for each candidate is defined as

$$\widehat{A}_i = \frac{r_i - mean(\{r_i\}_{i=1}^{G})}{std(\{r_i\}_{i=1}^{G})}. \tag{1}$$

This group-based normalization can be interpreted as a form of reward shaping: by emphasizing relative differences among candidate sequences for the same input, GRPO strengthens the reliability of the gradient signal. Instead of embedding a KL penalty inside the reward, GRPO directly regularizes the policy by including an explicit KL term between the learned policy and a reference policy. The

training objective is

$$\mathcal{J}_{GRPO}(\theta) = \mathbb{E}_{q \sim P(\mathcal{Q}), \{o_i\}_{i=1}^{G} \sim \pi_{\theta_{old}}(O|q)}$$

$$\frac{1}{G} \sum_{i=1}^{G} \frac{1}{|o_i|} \sum_{t=1}^{|o_i|} \min(r_{i,t}(\theta)\widehat{A}_{i,t}, clip(r_{i,t}(\theta), 1-\epsilon, 1+\epsilon)\widehat{A}_{i,t}) - \beta D_{KL}[\pi_\theta \| \pi_{ref}] \quad (2)$$

where $r_{i,t}(\theta) = \frac{\pi_\theta(o_{i,t}|q,o_{i,<t})}{\pi_{\theta_{old}}(o_{i,t}|q,o_{i,<t})}$ is the token-level importance ratio between the new and old policies, $\epsilon$ is the clipping range that prevents overly aggressive updates, $\beta$ controls the KL regularization strength, and $\pi_{ref}$ is a fixed reference policy. Here, $q$ denotes the input prompt, $o_i$ a generated sequence, and $o_{i,t}$ its $t$-th token.

**Limitations of GRPO.** Despite its popularity, GRPO inherits structural drawbacks from the underlying on-policy update rule. At each iteration, the parameters are updated via a **single stochastic gradient step** estimated from a batch of rollouts. The randomness of rewards leads to high-variance gradient estimates, making updates unstable. Group normalization partially dampens this effect but remains sensitive to fluctuations within each batch. Moreover, restricting each batch to a single update discards potentially useful gradient information across inner steps, leading to inefficient data use and limited variance reduction. These drawbacks underscore the need for an update mechanism that can stabilize gradient directions while making more effective use of available samples.

## 3 METHOD

To address the instability and inefficiency of one-shot policy updates, we propose **Slow-Fast Policy Optimization (SFPO)**, a simple yet general update mechanism that reuses rollouts more effectively while remaining plug-compatible with standard on-policy policy gradient algorithms. Each iteration consists of three stages: (i) a *fast trajectory* of multiple inner updates on the same batch, (ii) a *reposition* step that interpolates back toward the on-policy point to control drift, and (iii) a *slow correction* via an extra-gradient update. As illustrated in Fig. 1 and Alg. 1, this fast–reposition–slow design transforms noisy one-shot updates into well-structured update trajectories, yielding more stable optimization and higher sample efficiency without additional rollouts.

**Notation.** To align with standard gradient-based optimization notation, we define $\mathcal{L}(\theta) = -\mathcal{J}(\theta)$, so that maximizing the objective $\mathcal{J}$ is equivalent to minimizing the loss $\mathcal{L}$.

### 3.1 STAGE I: FAST TRAJECTORY

In standard on-policy policy-gradient methods such as GRPO, each outer iteration is updated by a single stochastic gradient:

$$\theta^{s+1} = \theta^s - \eta \nabla_\theta \mathcal{L}(\theta^s), \quad (3)$$

where $\nabla_\theta \mathcal{L}(\theta^s)$ is estimated from one batch of rollouts. Such one-shot updates suffer from high variance and often drive the policy in unstable directions, especially during early training. SFPO mitigates this by performing multiple inner updates on the *same* batch or rollouts.

Formally, starting from parameters $\theta^{s,0}$ at the beginning of iteration $s$, we execute a short *fast trajectory* of $K$ inner updates:

$$\theta^{s,k+1} = \theta^{s,k} - \eta \nabla_\theta \mathcal{L}(\theta^{s,k}), \quad k = 0, \dots, K-1. \quad (4)$$

This produces a sequence $\theta^{s,0} \to \theta^{s,1} \to \cdots \to \theta^{s,K}$, where each step refines the gradient direction using the same rollout data.

**Intuition.** Unlike one-shot updates that rely on a single noisy gradient, the displacement

$$\theta^{s,K} - \theta^{s,0} = -\eta \sum_{k=0}^{K-1} \nabla_\theta \mathcal{L}(\theta^{s,k}) \quad (5)$$

captures the cumulative effect of $K$ sequential corrections. Even if some individual steps are perturbed by noise, their composition tends to damp idiosyncratic fluctuations and align with the underlying

gradient direction. Geometrically, this can be viewed as integrating the local gradient field along a short trajectory in parameter space rather than trusting a single noisy vector at $\theta^{s,0}$. As a result, the update direction at the end of Stage I is typically more stable and less sensitive to randomness in any single gradient estimate.

**Compact Mathematical Intuition.** Let $\theta_k := \theta^{s,k}$ and consider the $K$ inner steps $\theta_{k+1} = \theta_k - \eta\nabla\mathcal{L}(\theta_k)$ starting at $\theta_0 = \theta^{s,0}$. In a small neighborhood, linearizing the gradient field, $\nabla\mathcal{L}(\theta) \approx g_0 + H_0(\theta - \theta_0)$ with $g_0 = \nabla\mathcal{L}(\theta_0)$ and $H_0 = \nabla^2\mathcal{L}(\theta_0)$, yields the closed-form displacement

$$\theta_K - \theta_0 \approx -\left[I - (I - \eta H_0)^K\right] H_0^\dagger g_0, \tag{6}$$

where $H_0^\dagger$ denotes the (pseudo)inverse on the range of $H_0$ (the $\lambda \to 0$ case is understood by continuity). Spectrally, along an eigen-direction with curvature $\lambda \geq 0$, the scalar gain is $\left(1 - (1 - \eta\lambda)^K\right)/\lambda$: for small $\lambda$ it behaves like $K\eta$ (steadily accumulating progress in gentle directions), while for larger $\lambda$ it saturates below 1 (damping stiff-direction oscillations). Thus the fast trajectory acts as a curvature-aware low-pass filter that stabilizes the endpoint direction relative to a one-shot step. *We use this as local intuition under a sufficiently small step size (e.g., $\eta < 1/\|H_0\|_2$) and in neighborhoods where positive-curvature directions dominate; in practice, KL/clip regularization and Stage II reposition mitigate adverse effects of negative curvature.*

## 3.2 STAGE II: REPOSITION

While the fast trajectory of Stage I improves stability, it also *changes the nature of the update from on-policy to off-policy*. Since all inner steps $\theta^{s,1},\ldots,\theta^{s,K}$ reuse the same rollouts generated at $\theta^{s,0}$, the endpoint $\theta^{s,K}$ no longer corresponds to the distribution that produced those samples. This *distribution mismatch* is a fundamental drawback of off-policy learning, as it biases gradient estimates and can destabilize training.

Inspired by Lookahead Optimization (Zhang et al., 2019), SFPO introduces a *reposition step* that interpolates the fast trajectory back toward its starting point:

$$\widetilde{\theta}^{s,K} = \theta^{s,0} + \alpha(\theta^{s,K} - \theta^{s,0}), \qquad \alpha \in [0,1]. \tag{7}$$

Here $\alpha$ regulates the degree of off-policy drift: smaller values keep the update close to the original on-policy iterate, while larger values rely more on the fast trajectory at the risk of greater mismatch.

**Intuition.** The interpolation is equivalent to solving a linearized proximal subproblem around $\theta^{s,0}$:

$$\min_\theta \left\langle \sum_{k=0}^{K-1} \nabla_\theta\mathcal{L}(\theta^{s,k}),\ \theta - \theta^{s,0} \right\rangle + \tfrac{\lambda}{2}\|\theta - \theta^{s,0}\|^2, \tag{8}$$

whose unique solution coincides with $\widetilde{\theta}^{s,K}$ for $\lambda = \frac{1}{\alpha\eta}$. Thus, $\alpha$ acts as an *implicit trust-region radius*: smaller $\alpha$ implies a larger proximal weight $\lambda$, enforcing stronger contraction toward the on-policy point. [2]

## 3.3 STAGE III: SLOW CORRECTION

After repositioning, SFPO applies one more (slow) correction step at the interpolated point:

$$\theta^{s+1} = \widetilde{\theta}^{s,K} - \eta\nabla_\theta\mathcal{L}(\widetilde{\theta}^{s,K}). \tag{9}$$

This yields a *predictor—corrector* structure: Stage I produces a stabilized *fast trajectory*, Stage II tempers off-policy drift via *reposition*, and Stage III applies a *slow correction* aligned with the local curvature at the update point.

---

[2] A heuristic alternative is obtained if one replaces $\sum_k \nabla_\theta\mathcal{L}(\theta^{s,k})$ by the averaged gradient $\bar{g}$ and uses the approximation $\theta^{s,K} - \theta^{s,0} \approx -\eta K\,\bar{g}$, leading to $\lambda \approx 1/(\alpha\eta K)$.

---

**Algorithm 1** SFPO: unified fast–reposition–slow update.

---

**Require:** Initial policy $\pi_{\theta^{0,0}}$, dataset $\mathcal{D}$, hyperparameters $S, K, \alpha_0, \eta, \omega, \tau$, loss $\mathcal{L}(\theta)$
    Initialize rolling buffer $\mathcal{H} \leftarrow \emptyset$, stats $(\mu, \sigma) \leftarrow (0, 1)$, trigger index $s^\star \leftarrow +\infty$
    **for** $s = 0, 1, \ldots, S - 1$ **do**
        Generate rollouts with the current policy $\pi_{\theta^{s,0}}$ on prompts from $\mathcal{D}$.
        $\alpha \leftarrow \alpha_0 \cdot \mathbf{1}[\, s < s^\star \,]$    *// Set $\alpha$ for this iteration from past trigger (non-anticipatory)*
        **if** $\alpha = 0$ **then**
            $\widetilde{\theta}^{s,K} \leftarrow \theta^{s,0}$    *// Skip fast trajectory & reposition*
        **else**
            **for** $k = 0, 1, \ldots, K - 1$ **do**
                $\theta^{s,k+1} \leftarrow \theta^{s,k} - \eta \nabla_\theta \mathcal{L}(\theta^{s,k})$    *// Stage I: Fast Trajectory*
            **end for**
            $\widetilde{\theta}^{s,K} \leftarrow \theta^{s,0} + \alpha(\theta^{s,K} - \theta^{s,0})$    *// Stage II: Reposition*
        **end if**
        $\theta^{s+1,0} \leftarrow \widetilde{\theta}^{s,K} - \eta \nabla_\theta \mathcal{L}(\widetilde{\theta}^{s,K})$    *// Stage III: Slow Correction*
        Compute entropy $H_s$; update rolling buffer $\mathcal{H}$ (keep last $\omega$ ones) and $(\mu_s, \sigma_s)$.
        $Z_s \leftarrow \frac{H_s - \mu_s}{\sigma_s + \varepsilon}$    ($\varepsilon$ *for numerical stability*)
        **if** $s^\star = +\infty$ **and** $|Z_s| \geq \tau$ **then**
            $s^\star \leftarrow s + 1$    *// will set $\alpha = 0$ for all future $s' \geq s^\star$*
        **end if**
    **end for**
    **return** final policy $\pi_{\theta^{S,0}}$

---

**Theoretical intuition.** Under $L$-smoothness and sufficiently small $\eta$, the descent lemma implies

$$\mathbb{E}[\mathcal{L}(\theta^{s+1})] \leq \mathcal{L}(\theta^{s,0}) - c\,\eta\|\nabla\mathcal{L}(\theta^{s,0})\|^2 + O(\eta^2 L \cdot \mathcal{F}(K, \alpha)), \tag{10}$$

where $\mathcal{F}(K, \alpha)$ represents the combined effect of Stage I (fast trajectory length $K$) and Stage II (reposition factor $\alpha$). Intuitively, $\mathcal{F}(K, \alpha)$ reflects a balance between exploiting more gradient information and controlling distributional drift: increasing $K$ leverages the same rollout data across multiple steps but also amplifies off-policy mismatch, while larger $\alpha$ interpolates more aggressively toward the fast trajectory at the risk of greater instability. In practice, since increasing $K$ raises both wall-clock cost and $\mathcal{F}(K, \alpha)$, we adopt small $K$ with moderately large $\alpha$, while the slow correction in Stage III mitigates overshooting and preserves progress along the stabilized trajectory direction.

### 3.4 Scheduling $\alpha$

A nonzero $\alpha$ is essential to exploit the stabilized *fast trajectory* when $K > 0$: if $\alpha = 0$, the reposition collapses to $\theta^{s,0}$ and the fast trajectory is discarded, eliminating the benefit of Stage I. However, the same aggressiveness that helps early can be counter-productive near a minimizer. When $\|\nabla\mathcal{L}\|$ is large, a nonzero $\alpha$ accelerates progress by moving along the stabilized fast direction; but as we approach a minimum, the signal weakens while curvature and stochastic noise dominate, so a large $\alpha$ amplifies drift and instability. This motivates an *adaptive schedule for $\alpha$*.

**Why adapt $\alpha$ rather than $K$?** $K$ controls both stabilization and wall-clock runtime: increasing $K$ reduces oscillations but incurs proportional compute cost, making mid-training changes impractical. By contrast, $\alpha$ is a *soft trust parameter*: it decides how much of the stabilized fast trajectory is exploited, without changing runtime or discarding the already-computed $K$ steps. Thus $\alpha$ is the natural lever to adapt dynamically.

**Adaptive rule in practice.** In our implementation, $\alpha$ is scheduled online at each iteration. After generating rollouts with the current policy, we compute the policy entropy $H_s$ and maintain a rolling buffer of the past $\omega$ entropy values (window size $\omega$). Let $\mu_s$ and $\sigma_s$ denote the mean and standard deviation within this buffer. We define the one-sided z-score

$$Z_s = \frac{H_s - \mu_s}{\sigma_s}. \tag{11}$$

If $|Z_s| \geq \tau$ for a threshold $\tau$, we mark the current step $s^\star$ and set $\alpha = 0$ for all $s \geq s^\star$. Otherwise we keep $\alpha = \alpha_0$. Intuitively, sharp entropy fluctuations signal that the policy is close to a local optimum

where noise dominates, so interpolation should be disabled. This entropy-triggered schedule exploits the fast trajectory in the high-signal early phase, while reverting to pure on-policy updates near convergence for stability.[3]

## 3.5 UNIFIED SFPO UPDATE

Collecting the three stages, the unified SFPO update for each step $s$ is:

$$\theta^{s+1} = \theta^{s,0} - \eta\Big[ \underbrace{\alpha \sum_{k=0}^{K-1} \nabla_\theta \mathcal{L}(\theta^{s,k})}_{\text{fast trajectory \& reposition}} + \underbrace{\nabla_\theta \mathcal{L}(\widetilde{\theta}^{s,K})}_{\text{slow correction}} \Big], \tag{12}$$

As shown in Alg. 1, SFPO unifies the three stages into a single update rule and serves as a plug-compatible drop-in replacement for on-policy policy-gradient methods such as GRPO. Our experiments show that this structural change consistently improves stability and sample efficiency on diverse math reasoning benchmarks, with minimal engineering overhead.

## 4 EXPERIMENTS

### 4.1 EXPERIMENTAL SETTINGS

**Models.** We conduct the reasoning rl training with our proposed SFPO on a wide range of models: Qwen2.5-Math-1.5B (Yang et al., 2024), DeepSeek-R1-Distill-Qwen-1.5B (DeepSeek-AI et al., 2025a), Qwen3-4B-Base (Yang et al., 2025), Qwen2.5-Math-7B (Yang et al., 2024), and DeepSeek-R1-Distill-Qwen-7B (DeepSeek-AI et al., 2025a). For Qwen2.5-Math-1.5B/7B, we set the context length in the training process as 4096 the same as their maximum context length; while for Qwen3-4B-Base and DeepSeek-R1-Distill-Qwen-1.5B/7B, we set the context length as 8192 for the better accuracy-efficiency tradeoff.

**Reasoning RL Training.** We conduct the reasoning RL training on two different training datasets to test the effectiveness of SFPO on different scales of dataset. The first one is the combination of DAPO training dataset (Yu et al., 2025a) and Math training dataset (Hendrycks et al., 2021), which is a total of approximate 24K data. The second one is the Skywork-OR1 Math RL training dataset (He et al., 2025b) with 105k data. The training batch size is set to 256, and the number of responses for each question is set to 8 by default. The total trianing step is set to 400 by default. All the training experiments are done based on verl (Sheng et al., 2025) with a single 8-GPU node.

**Baseline and Evaluation.** We compare SFPO with vanilla GRPO and the base model without rl training on six commonly used mathematical reasoning benchmarks with variant difficulty: Math500 (Hendrycks et al., 2021), AIME24 (Art of Problem Solving, 2024a), AIME25 (MAA), AMC (Art of Problem Solving, 2024b), MinervaMath (Lewkowycz et al., 2022), Olympiad Bench (He et al., 2024). Each benchmark is evaluated multiple times with rollout temperature being 1, and we report the average Pass@1 accuracy by default.

### 4.2 MAIN RESULTS

#### 4.2.1 MATH REASONING BENCHMARKS

As illustrated in Table 1, our proposed SFPO consistently outperforms vanilla GRPO across all base models and benchmarks. Specifically, for small-scale models such as Qwen2.5-Math-1.5B and DS-distilled-Qwen-1.5B, SFPO demonstrates superior performance enhancements on math reasoning benchmarks, raising the average accuracy from 38.35 to 40.19 with an absolute gain of **+1.84**, and from 47.73 to 50.53 with a gain of **+2.80**, respectively. The improvements are particularly pronounced on challenging tasks such as AIME24 and AIME25, where DS-distilled-Qwen-1.5B achieves an absolute gain of **+7.5** on AIME25. The larger models also exhibit similar performance

---

[3]Although a decaying schedule on $\alpha$ could be a formal solution, we find little empirical difference, as shown in Fig. 7 of Sec. 5. Hence, for simplicity and efficiency, we downgrade SFPO to GRPO once the iteration reaches $s \geq s^\star$.

Table 1: Performance on math reasoning benchmarks with DAPO and Math training dataset.

| Model | Method | Math-500 | AIME24 | AIME25 | AMC | Minerva | Olympiad | Avg |
|---|---|---|---|---|---|---|---|---|
| **Qwen2.5-Math-1.5B** | Base | 55.55 | 9.17 | 5.83 | 37.65 | 17.74 | 28.45 | 25.73 |
| | GRPO | 77.15 | 16.67 | 11.67 | 53.31 | 31.89 | 39.42 | 38.35 |
| | SFPO | **78.35** | **20.00** | **15.00** | **56.02** | **32.07** | **39.72** | **40.19** |
| **Qwen2.5-Math-7B** | Base | 71.65 | 21.67 | 9.17 | 53.61 | 27.02 | 38.54 | 36.94 |
| | GRPO | 82.50 | 34.20 | 20.83 | 71.08 | 36.76 | 44.80 | 48.36 |
| | SFPO | **82.30** | **35.00** | **20.83** | **74.49** | **36.94** | **45.59** | **49.19** |
| **DS-distilled-Qwen-1.5B** | Base | 73.80 | 16.67 | 20.00 | 47.59 | 28.86 | 36.94 | 37.31 |
| | GRPO | 84.65 | 30.00 | 23.33 | 66.86 | 31.71 | 49.85 | 47.73 |
| | SFPO | **86.10** | **32.50** | **30.83** | **70.28** | **32.81** | **50.67** | **50.53** |
| **DS-distilled-Qwen-7B** | Base | 83.60 | 28.33 | 25.00 | 61.75 | 41.73 | 48.89 | 48.21 |
| | GRPO | 91.7 | 50.00 | 35.83 | 80.42 | 43.65 | 61.24 | 60.47 |
| | SFPO | **92.60** | **54.17** | **37.50** | **83.75** | **44.49** | **65.73** | **63.04** |
| **Qwen3-4B-Base** | Base | 45.25 | 2.50 | 0.83 | 20.48 | 15.99 | 20.66 | 17.62 |
| | GRPO | 83.35 | 16.67 | 17.50 | **59.03** | 38.78 | 48.59 | 43.99 |
| | SFPO | **84.30** | **21.67** | **20.83** | 57.23 | **40.81** | **48.67** | **45.59** |

gains. For Qwen2.5-Math-7B, SFPO raises the average accuracy from 48.36 to 49.19 with an absolute gain of **+0.83**. For DS-distilled-Qwen-7B, SFPO boosts the average accuracy from 60.47 to 63.04, corresponding to an absolute gain of **+2.57**. For Qwen3-4B-Base model, SFPO improves average accuracy from 43.99 to 45.59, an absolute gain of **+1.60**, highlighting its robustness across various models. Moreover, Table 2 demonstrates that SFPO can consistently achieve better training performance compared to GRPO for the larger training dataset Skywork-or1, proving the robustness of SFPO across different scales and distributions of training datasets.

### 4.2.2 TRAINING DYNAMICS.

We compare the training dynamics between SFPO and GRPO to better understand the differences in optimization behavior as shown in Fig. 2 and Fig. 3.

**Validation.** From Fig. 2, we can clearly spot that SFPO consistently outperforms GRPO across all base models throughout the training process. Not only does SFPO achieve faster convergence in the early stages, but it also sustains higher global accuracy by the end of training. For example, Qwen3-4B-Base model achieves a sharper rise and stabilizes at a higher accuracy within only 150 training steps, while vanilla GRPO cannot surpass this accuracy even after 400 steps.

Table 2: Performance on AIME24/25 with Skywork-or1 training dataset.

| Model | Method | AIME24 | AIME25 |
|---|---|---|---|
| **DS-Qwen-1.5B** | Base | 20.40 | 17.90 |
| | GRPO | 32.92 | 25.83 |
| | SFPO | **34.17** | **27.50** |
| **DS-Qwen-7B** | Base | 25.42 | 22.92 |
| | GRPO | 42.50 | 29.20 |
| | SFPO | **43.75** | **30.00** |

**Response Length.** Moreover, distinct training behaviors between SFPO and GRPO for DeepSeek-R1-Distill-Qwen-7B are shown in Fig. 3 including the response length, entropy loss, and the reward throughout the training process.

Specifically, GRPO gradually collapses to overly short responses while SFPO quickly converges to a stable range of around 2700 tokens with better accuracy, highlighting SFPO's ability to regulate response length more effectively and avoid overthinking with verbose responses (Liu et al., 2025).

**Entropy.** From Fig. 3(b), we can observe that SFPO makes model's entropy loss lower compared to GRPO. Typically, lower entropy means weak exploration ability for reasoning models; however, the entropy reduction under SFPO mainly reflects the model's ability to eliminate unproductive search paths early, rather than suppressing exploration altogether as evidenced by its sustained accuracy gains. In fact, the model still explores a sufficiently broad set of reasoning trajectories; therefore, the lower entropy observed under SFPO should be viewed as a sign of more efficient exploration rather than a sign of limited exploration.

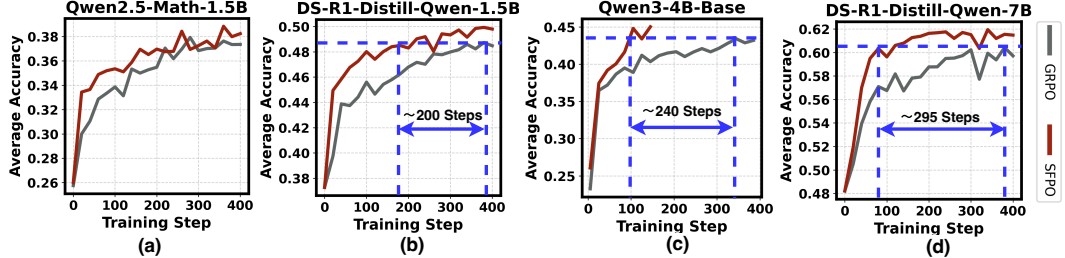

Figure 2: Average validation accuracy of different base models throughout the learning process.

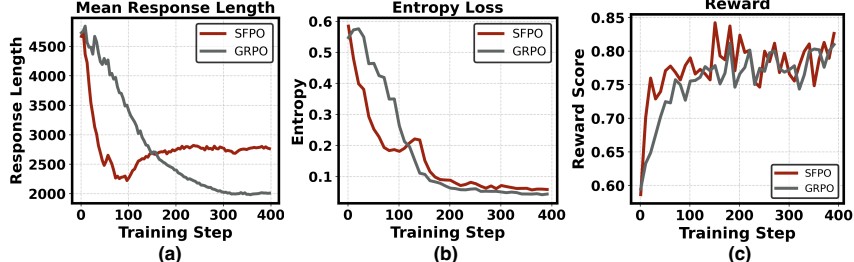

Figure 3: Comparison of training behaviors in terms of response length, entropy loss, and reward.

**Reward Score.** SFPO also achieves a higher and more stable reward throughout the training process compared to GRPO, indicating stronger alignment with the reward function and more robust convergence. This is further reflected in its superior accuracy and well-controlled response length.

### 4.2.3 EFFICIENCY ANALYSIS.

We evaluate the efficiency gains of SFPO over GRPO by comparing the total number of rollouts and the wall-clock time required to reach the same benchmark accuracy. The results in Fig. 4 illustrate that SFPO consistently outperforms GRPO in both rollout efficiency and training time across all model scales. To be specific, SFPO requires 3.21×, 3.50×, and 4.93× fewer rollouts than GRPO for DS-Qwen-1.5B, Qwen3-4B-Base, and DS-Qwen-7B, respectively, to reach the same best accuracy. This advantage directly translates into reduced training time, where SFPO

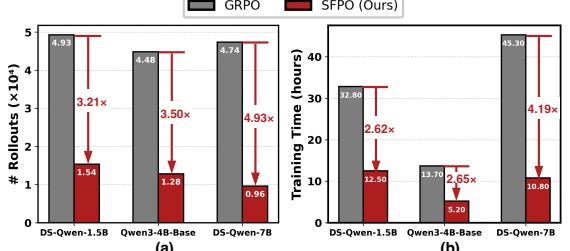

Figure 4: Comparison between GRPO and SFPO. (a) Number of rollouts required to achieve the same best accuracy as GRPO. (b) Corresponding training time.

achieves 2.62×, 2.65×, and 4.19× speedups over GRPO for the same models, significantly lowering the training cost. Note that SFPO does not introduce extra GPU memory overhead as it does not need to store the heavy optimizer status. The detailed profiling results for GPU memory usage in the training process can be found in Appendix B. These significant efficiency gains align with our expectations, since the primary bottleneck in the training process lies in rollout generation, which accounts for more than 70% of the overall inference time (He et al., 2025a). By substantially reducing the number of rollouts required and harnessing the reposition mechanism, SFPO alleviates this bottleneck and achieves faster training.

## 5 ANALYSIS AND ABLATION STUDY

**Impact of $\alpha$ and $K$.** As discussed in Sec. 3, the hyperparameters $\alpha$ and $K$ jointly determine the stability of SFPO. To better understand their impacts, we study two settings: small $K{=}3$ and large $K{=}7$, under varying $\alpha$, as shown in Fig. 5. When $K$ is small, SFPO remains stable across different $\alpha$ values and consistently outperforms GRPO. This aligns with our theoretical intuition that performing multiple inner updates on the same rollout batch, followed by reposition, effectively reduces gradient

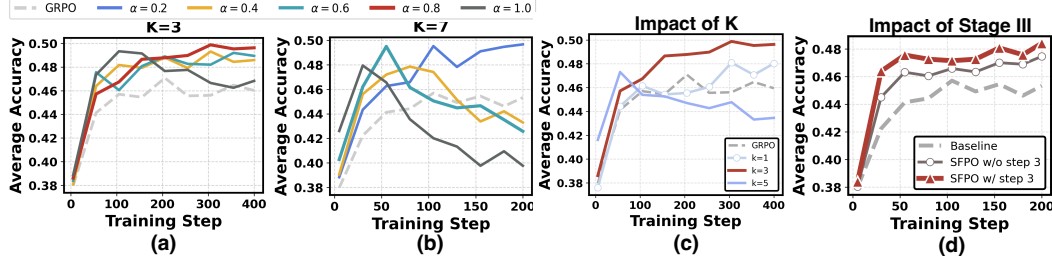

Figure 5: Average training accuracy of different settings throughout the training process. (a): Small k=3 with varying values of $\alpha$. (b): Large k=7 with varying values of $\alpha$. (c): Varying values of k with fixed $\alpha = 0.8$. (d): Impact of the existence of stage III.

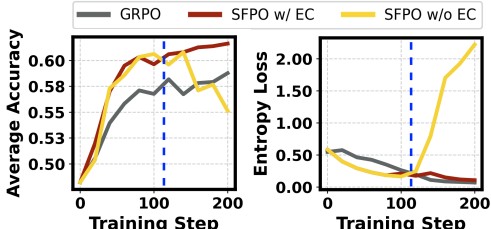

Figure 6: Comparison between SFPO w/ and w/o entropy control (EC). The blue dashed line indicates the stop step identified by z-score.

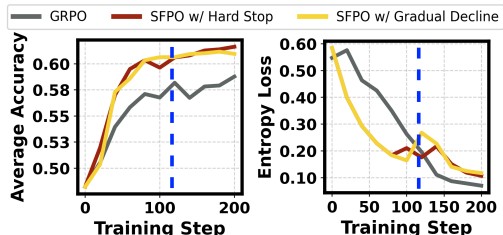

Figure 7: Comparison between SFPO with different $\alpha$ decay strategies. The blue dashed line indicates the stop step identified by z-score.

noise and produces a more reliable update direction. However, when $K$ is large, the fast weights drift substantially from the original parameters. A large $\alpha$ then amplifies this mismatch by pulling the slow weights too aggressively toward unstable fast trajectories, injecting noise and causing performance drops. As shown in Fig. 5(b), a smaller $\alpha$ mitigates this drift and restores stability, confirming the interaction between $K$ and $\alpha$.

**Interpolation Against Off-Policy Overfitting.** When $\alpha = 1$ (no interpolation), performance initially rises but steadily declines as training progresses. Without interpolation, the model rapidly adapts by overfitting to small batches of rollouts, yielding short-term gains but injecting growing noise into gradient updates, which causes instability and long-term degradation. This effect is amplified when $K$ is large, as shown in Fig. 5(b), where the fast weights drift substantially and the slow weights fully adopt these noisy trajectories, leading to sharp performance drops. In contrast, for small $\alpha$, interpolation keeps the update closer to the on-policy region, mitigating distributional drift and stabilizing training. However, if $\alpha$ is overly small and $K$ is small, the algorithm under-utilizes the stabilized fast trajectory, slightly slowing early improvements. While for large $K$, a smaller $\alpha$ is preferable, where interpolation effectively counteracts the severe off-policy drift induced by many inner steps. Overall, interpolation serves as a simple, effective regularizer against the potential off-policy overfitting introduced by Stage I.

**The Importance of Stage III: Slow Correction.** As shown in Fig. 5(d), incorporating slow correction consistently improves stability and accuracy over GRPO. Without this stage, the reposition stage leaves the iterate at an interpolated point that may deviate from the true descent direction. Intuitively, slow correction provides a curvature-aware adjustment that realigns updates with the correct optimization trajectory. Beyond its immediate benefit, slow correction also provides a natural interface for future extensions: As discussed in Sec. 7, one could, for instance, perform an additional rollout on a small held-out meta-test batch after repositioning, enabling curriculum or meta-learning variants of SFPO without altering its overall structure.

**Necessity of Entropy Control.** We evaluate the effect of adaptive entropy control (EC) on SFPO using the DS-Qwen-7B model. As shown in Fig. 6, removing EC causes a noticeable accuracy drop after roughly 100 steps. This degradation coincides with a rapid divergence of entropy loss, suggesting that the policy becomes unstable and overfits to noisy rollouts. These results highlight entropy control as a key factor for maintaining the stability and reliability of SFPO. For simplicity, we implement entropy control through a lightweight scheduling of $\alpha$. As shown in Alg. 1, once a

predefined entropy-trigger is activated, $\alpha$ is set to $0$, reverting SFPO to standard GRPO for subsequent steps. This simple mechanism effectively stabilizes training while retaining the efficiency gains.

**Strategies for Decaying $\alpha$.** We further investigate alternative strategies for reducing the value of $\alpha$ once the stop step is identified by the z-score criterion. Specifically, we compare two representative approaches: (i) our default method, where $\alpha$ is directly set to zero after the trigger step $s^\star$, and (ii) a more gradual linear decay schedule, where $\alpha$ decreases to zero over several subsequent steps. As illustrated in Fig. 7, both strategies yield similar accuracy and stability curves, suggesting that the decay schedule itself has negligible influence on the overall performance of SFPO. This result implies that once the entropy-trigger condition is met, the role of $\alpha$ becomes marginal, and any further adjustment has limited benefit. Consequently, for simplicity and wall-clock efficiency, we adopt the direct reset rule of setting $\alpha = 0$ after $s^\star$. The formal procedure is summarized in Alg. 1, and additional details can be found in Sec. 3.4.

# 6 RELATED WORKS

**RL for LLM Reasoning.** OpenAI O1 (OpenAI) introduced a paradigm shift in LLM reasoning by extending the reasoning horizon before final responses. DeepSeek-R1 (Guo et al., 2025) further advanced this line by open-sourcing both its training algorithm, the value-model-free GRPO (Shao et al., 2024), and model weights, achieving performance comparable to O1. Subsequent work has focused on stabilizing and simplifying GRPO: DAPO (Yu et al., 2025b) identifies entropy collapse as a key challenge and proposes effective remedies, while Dr. GRPO (Liu et al., 2025) removes normalization terms without sacrificing performance. In contrast, SFPO is orthogonal to both families: it leaves the objective unchanged yet restructures the update itself into a fast–reposition–slow decomposition, improving variance reduction, stability, and sample efficiency in LLM reasoning.

**Data Efficiency for LLM Reasoning.** Despite focusing on designing novel training pipelines, a complementary line of work (Ivison et al., 2025; Xia et al., 2024; Muennighoff et al., 2025; Ye et al., 2025) improves the efficiency of LLM training through data filtering. One direction focuses on pruning data for supervised fine-tuning (Xia et al., 2024; Chen et al., 2023; Ivison et al., 2022). Another direction targets reinforcement learning, where studies (Muldrew et al., 2024; Liu et al., 2024; Das et al., 2024; Li et al., 2025; Fatemi et al., 2025; Wang et al., 2025) show that GRPO requires only a small subset of the training data to improve reasoning ability. However, these methods largely optimize *which data* is used rather than *how updates are performed*. SFPO is orthogonal: it assumes no change to the data pipeline but restructures the policy update itself. By converting one-shot updates into a fast–reposition–slow trajectory, SFPO reduces variance and stabilizes learning.

**Off-policy GRPO.** Original iterative GRPO (Shao et al., 2024) tries to update policy model for multiple times with the same batch of rollout data. Mroueh (2025) provided a theoretical analysis for this off-policy policy gradient operation under verifiable rewards, reformulating GRPO as a KL-regularized contrastive loss and proving its iterative dynamics amplify success rates relative to the reference policy. Mroueh et al. (2025) examined both on- and off-policy variants, deriving reward improvement conditions, introducing clipped surrogate objectives to stabilize training, and empirically showing off-policy GRPO can match or exceed on-policy performance. Unlike these modifications, SFPO preserves the on-policy GRPO update but restructures one-step optimization into a fast-reposition-slow framework to tackle sample inefficiency and stabilize updates.

# 7 CONCLUSION

We proposed **Slow-Fast Policy Optimization (SFPO)**, a simple three-stage update mechanism that stabilizes early training by combining a fast trajectory, a reposition step, and a slow correction. SFPO directly addresses the instability of one-shot GRPO updates, achieving higher reasoning accuracy and fewer generated tokens without added wall-clock cost. Beyond these gains, its modular structure naturally opens avenues for curriculum or meta-learning extensions, and our entropy-triggered $\alpha$ schedule suggests richer adaptive rules. We believe SFPO offers not only an effective plug-in for current policy optimization, but also a foundation for scalable and data-aware optimization strategies in the LLM reasoning area.

## ACKNOWLEDGMENTS

Huo is partially sponsored by a subcontract of NSF grant 2229876, the A. Russell Chandler III Professorship at Georgia Institute of Technology, an NIH-sponsored Georgia Clinical & Translational Science Alliance, and the Georgia Department of Transportation.

## REPRODUCIBILITY STATEMENT

All results reported in Sec. 4 are fully reproducible. We will release the code and experiment scripts upon acceptance.

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

## A    LLM USAGE

Large language models (LLMs) were used solely to improve the writing of this paper, including grammar, clarity, and readability. They were not used for generating ideas, designing experiments, conducting analyses, or producing scientific content. All research contributions, technical claims, and conclusions are entirely the work of the authors.

## B    GPU MEMORY PROFILING RESULTS

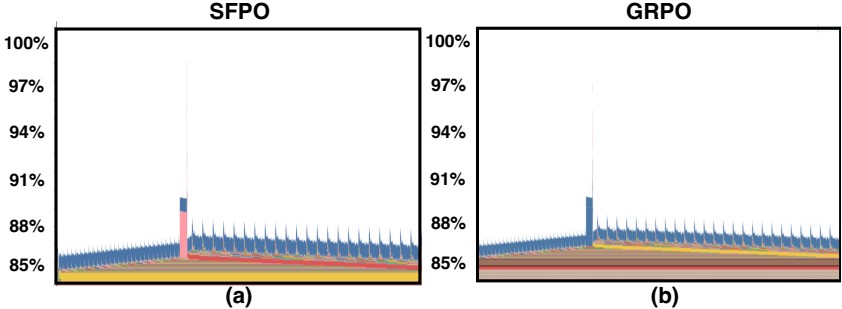

Figure 8: Comparison of GPU memory consumption during one RL training step between SFPO and GRPO for Deepseek-R1-Distill-Qwen-7B model.

From Fig. 8, we can clearly observe that SFPO and GRPO demonstrate similar GPU memory consumption during one training step. This aligns with our expectation: SFPO does not need to store the heavy optimizer states and parameters but only need to store one copy of the model weight, which does not introduce significant overhead, especially when the model is sharded across GPUs.

## C    ADDITIONAL RESULTS

### C.1    IMPACT OF INTERPOLATION SCHEME IN STAGE II

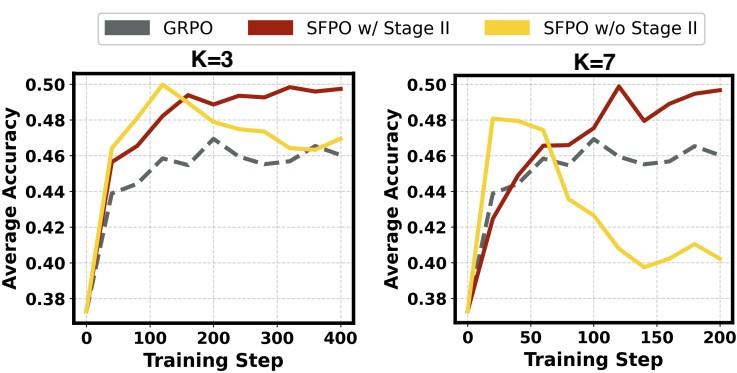

Figure 9: Ablation on interpolation scheme in Stage II with DeepSeek-R1-Distill-Qwen-1.5B.

To make the significance of stage II much clearer, we further conduct an explicit comparison between SFPO with and without the interpolation scheme in stage II using DeepSeek-R1-Distill-Qwen-1.5B, and the results are demonstrated in the Fig. 9. Specifically,

- When $K = 3$, SFPO without Stage II is slightly better at the very beginning of training because the drift accumulated over only three fast updates remains relatively small. However, as training progresses, its performance begins to degrade due to the growing off-policy mismatch. Since $K = 3$ is still relatively close to the vanilla GRPO, the degradation in the later stage becomes comparable to standard GRPO.

- When $K = 7$, the difference becomes substantially more pronounced. SFPO without Stage II initially gains some accuracy, but the lack of a reposition step causes severe off-policy drift as training continues. With $K = 7$, this drift grows large enough that the method moves far away from the on-policy region used by standard GRPO, eventually causing the training to collapse. In contrast, SFPO with Stage II remains stable throughout training and consistently achieves the best overall performance.

These results confirm that the interpolation-based reposition step in Stage II is crucial for stabilizing multi-step updates and turning the additional inner updates into consistent gains.

## C.2 SFPO TRAINING ON CODING TASKS

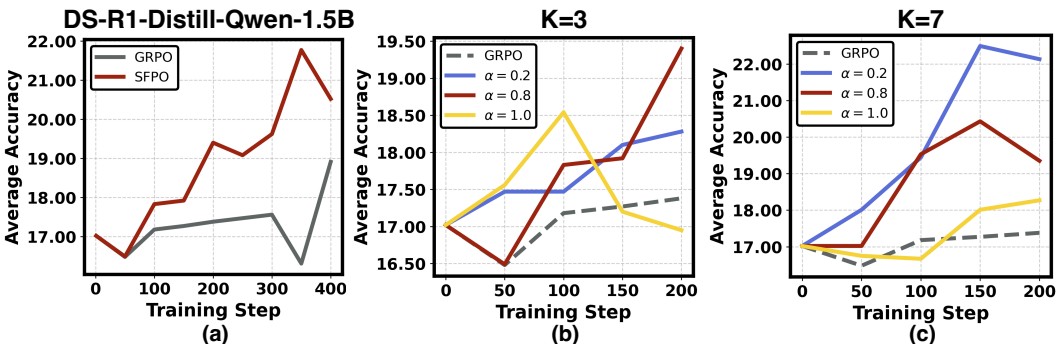

Figure 10: (a): Validation accuracy of LiveCodeBench on DeepSeek-R1-Distill-Qwen-1.5B between SFPO and GRPO throughout the learning process. (b): Ablation of $\alpha$ when $K = 3$ for coding tasks. (c): Ablation of $\alpha$ when $K = 7$ for coding tasks.

We additionally trained DeepSeek-R1-Distill-Qwen-1.5B on the Skywork-OR1 Code RL training dataset (He et al., 2025b) with 14.1k data. The training hyperparameters mirror those used for math tasks, with $K = 3$ and $\alpha = 0.8$ by default. We then evaluate the resulting checkpoints on LiveCodeBench with a 32K response length and report Pass@1 (Avg 4). The results are illustrated in Fig. 10(a). As shown, within the 400 training steps, GRPO reaches a maximum Pass@1 (Avg 4) of 18.91 at 400 steps, whereas SFPO attains a higher peak performance of 21.77 at 350 steps. This consistent improvement on the coding benchmark provides additional evidence that the efficacy of SFPO extends beyond purely mathematical reasoning tasks.

We also conduct an additional ablation study to investigate the impact of $\alpha$ and $K$ on RL training for coding tasks. Specifically, we consider two scenarios: (i) a small inner-update budget $K = 3$ with varying $\alpha \in 0.2, 0.8, 1.0$, and (ii) a larger budget $K = 7$ with the same set of $\alpha$ values. As shown in Fig. 10(b) and Fig. 10(c), when $K = 3$, the default value $\alpha = 0.8$ achieves the best accuracy through out the training process, which is consistent with the findings in math tasks. While for $K = 7$, a smaller $\alpha = 0.2$ is preferred to mitigate stronger distribution drift and restore stability, again aligning with the trends observed in math RL training.

## C.3 INTERPOLATION IN STAGE-II AGAINST KL PENALTY

We conducted additional experiments comparing the KL-penalty baseline with the interpolation scheme used in our SFPO method. Specifically, we use Qwen3-4B-Base as the policy model. The training dataset and all other settings are identical to those in Section 4.1 of the submitted manuscript. We then compare interpolation against the KL-penalty baseline over mathematical reasoning benchmarks under two inner update settings, with $K = 3, \alpha = 0.8$ and $K = 7, \alpha = 0.2$, respectively.

The results are shown in the Fig. 11. We can observe that for k=3, interpolation consistently outperforms the KL-penalty variant throughout training; for example, at step 200 interpolation reaches **45.44**, compared to **40.64** with KL penalty. The gap becomes even more pronounced when k=7: the interpolation scheme remains stable and continues to improve up to **46.14** at step 200, whereas the KL-penalty variant becomes unstable and collapses after step 80, yielding zero

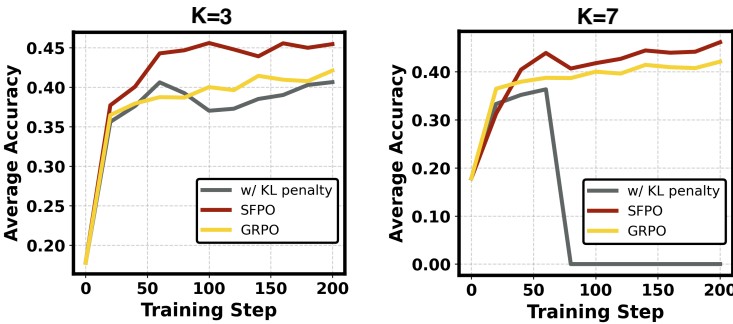

Figure 11: Comparison of using interpolation scheme in stage II (SFPO) and using KL divergence penalty in stage II on Mathematical reasoning benchmarks for Qwen3-4B-Base through the RL training process.

scores. These results empirically demonstrate our interpolation-based Stage II provides both better performance and substantially improved stability compared to a KL-constrained update.

## C.4 APPLYING SFPO ON DAPO

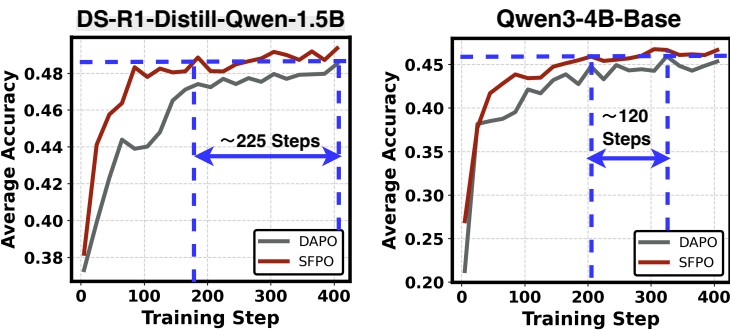

Figure 12: **Comparison of DAPO and SFPO on math benchmarks.** Validation average accuracy versus training step for DeepSeek-R1-Distill-Qwen-1.5B (left) and Qwen3-4B-Base (right).

To directly examine whether the performance gains of SFPO come primarily from handling zero-advantage batches, we compare DAPO against DAPO with SFPO applied using DeepSeek-R1-Distill-Qwen-1.5B and Qwen3-4B-Base as the policy models. The validation accuracy on the math benchmarks throughout the training process is reported in Fig. 12. From these two curves, we observe that for both models, SFPO not only achieves higher accuracy than DAPO over the entire training trajectory, but also reaches the target performance substantially earlier: roughly 225 steps faster on DS-R1-Distill-Qwen-1.5B and 120 steps faster on Qwen3-4B-Base. We further summarize the best accuracy achieved during training in the Table 3, where for each method we select the checkpoint with the highest average score across all math benchmarks. On DS-R1-Distill-Qwen-1.5B, SFPO attains an average accuracy of 50.56, while DAPO reaches 49.30; on Qwen3-4B-Base, SFPO achieves 47.87 compared to 46.48 for DAPO, with consistent improvements across most individual benchmarks.

Since DAPO already employs dynamic sampling to mitigate the zero-advantage issue, these results indicate that SFPO continues to provide clear benefits even in a setting where this particular source of variance is explicitly addressed. This suggests that SFPO's gains do not primarily stem from handling zero-advantage batches. Instead, SFPO operates at a **higher-level optimization layer** that is independent of the internal mechanics of GRPO or DAPO; it is therefore orthogonal to these methods and can be seamlessly integrated with GRPO, DAPO, or any GRPO-family variant.

## C.5 COMPUTE COST AND WALL-CLOCK EFFICIENCY OF SFPO VS GRPO

In SFPO, each iteration performs $K$ fast policy updates on the same batch, so the backward-pass compute for the policy update scales approximately linearly with $K$. Previous works have shown that under GRPO, about 70% of the per-step time is spent on rollout generation, $\approx 20\%$ on the policy update, and $\approx 10\%$ on other overhead. Increasing $K$ from 1 to 3 multiplies only the 20%

Table 3: Performance on math reasoning benchmarks for DAPO and applying SFPO on top of DAPO.

| Model | Method | Math-500 | AIME24 | AIME25 | AMC | Minerva | Olympiad | Avg |
|---|---|---|---|---|---|---|---|---|
| DS-distilled-Qwen-1.5B | DAPO | 85.40 | 32.50 | 24.17 | 68.07 | 33.64 | 52.00 | 49.30 |
| | SFPO | **86.25** | **33.33** | **26.67** | **70.18** | **34.38** | **52.52** | **50.56** |
| Qwen3-4B-Base | DAPO | 84.50 | 22.50 | 20.83 | **59.64** | 39.98 | 51.41 | 46.48 |
| | SFPO | **85.55** | **25.83** | **23.33** | 58.13 | **41.27** | **53.12** | **47.87** |

"policy-update" portion, so the worst-case theoretical overhead is

$$\frac{T_{\text{SFPO}}}{T_{\text{GRPO}}} \approx 1 + (K - 1) \cdot 0.2 = 1.4. \tag{13}$$

In practice, measured average per-step wall-clock times on DeepSeek-R1-Distill-Qwen-1.5B are $\sim 300\,\text{s}$ ($\sim 240\,\text{s}$ for rollout generation and $\sim 40\,\text{s}$ for the actor update) for GRPO and $\sim 410\,\text{s}$ ($\sim 240\,\text{s}$ for rollout generation and $\sim 150\,\text{s}$ for the actor update) for SFPO (with $K = 3$ and $\alpha = 0.8$), i.e., SFPO steps are only $\approx 1.37\times$ more expensive than GRPO steps. Moreover, our ablations on $\alpha$ and $K$ over both math and coding benchmarks show that this relatively small $K = 3$ with a moderately large $\alpha = 0.8$ is already sufficient; larger $K$ does not yield further gains and is therefore unnecessary in practice.

We compare SFPO and GRPO on DeepSeek-R1-Distill-Qwen-1.5B in terms of validation accuracy versus wall-clock training time in Fig. 13. Across essentially the entire range from about 5 to 33 hours, the SFPO curve stays above the GRPO curve, indicating that for any fixed time budget SFPO yields better policies. Quantitatively, SFPO is typically about 1–3.6 accuracy points higher than GRPO at matched wall-clock times, despite having slightly higher per-step compute. Conversely, if we fix accuracy targets, SFPO reaches the same accuracy level in 3–6.6× less wall-clock time, showing that its additional inner updates translate into substantially improved overall compute efficiency.

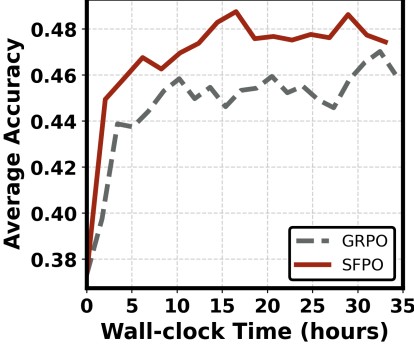

Figure 13: Wall-clock efficiency of SFPO vs. GRPO on DeepSeek-R1-Distill-Qwen-1.5B.

In summary, while Stage I of SFPO does incur additional gradient updates and thus increases per-iteration compute as $K$ grows, we deliberately operate in a regime with a small $K$ where the per-step overhead is modest. Under fair comparisons at matched wall-clock time and at fixed accuracy targets, SFPO is substantially more compute-efficient than GRPO, indicating that the extra Stage-I cost is more than compensated by improved sample and optimization efficiency.

## D DISCUSSION

### D.1 NOVELTY

**Three-stage training pipeline.** In the standard GRPO pipeline, the model first generates rollouts and then performs a single update using the GRPO objective. However, this one-shot update makes early training highly sample-inefficient and sensitive to noisy rollouts, increasing both training cost and instability. In contrast, our three-stage pipeline (fast → reposition → slow) reaches the best GRPO performance many steps earlier than vanilla GRPO. This results in a substantial reduction in rollout usage and overall training budget, while preserving the original GRPO objective and rollout process.

**A plug-in, higher-level optimization mechanism.** Our SFPO does not require substantial modifications to GRPO because it operates at a higher level of the optimization pipeline. Unlike GRPO variants that introduce different twists to advantage computation, SFPO is orthogonal to these methods and can be seamlessly combined with them without changing their training objective, KL regularization, or rollout procedure. This makes SFPO an easy-to-use, plug-and-play mechanism applicable to any GRPO-family method.

## D.2 NAMING OF SFPO

When naming our SFPO method, the terms "fast" in Stage I and "slow" in Stage III are simply naming choices. In our terminology, "slow" and "fast" refer to the frequency of updates, not the computational speed. Specifically, Stage I performs multiple fast, high-frequency updates (K inner steps), whereas Stage III performs only a single correction update. Thus, Stage I is called fast due to its update frequency, while Stage III is called slow because it occurs only once per iteration.

## D.3 MORE ENTROPY ANALYSIS

In general, lower entropy can sometimes indicate premature convergence to an over-deterministic policy. In our setting, however, the observed entropy reduction in SFPO is better interpreted as more efficient exploration, rather than collapsing exploration.

In vanilla GRPO, each update depends on a single-step, high-variance advantage estimate, which makes the early update direction unstable. As a result, GRPO often keeps higher entropy to compensate for this noise.

In contrast, SFPO aggregates multiple gradient steps (Stage 1), producing a much more stable and accurate update direction. Given this reduced variance, the policy can more confidently up-weight promising actions and down-weight clearly suboptimal ones, so it does not need to maintain as much randomness to keep improving. Importantly, SFPO still retains GRPO's entropy bonus, so it does not suppress exploration intentionally.

Overall, the lower entropy we observe corresponds to more efficient exploration, not reduced exploration. The performance improvements and stable diversity metrics strongly support this interpretation.

## D.4 INFORMAL DEFINITION AND ANALYSIS OF $\mathcal{F}(K, \alpha)$ IN SEC. 3.3.

We adopt a local quadratic-with-noise model around $\theta^{s,0}$: $\mathcal{L}(\theta) \approx \frac{1}{2}(\theta - \theta^\star)^\top H(\theta - \theta^\star)$ with $H \succeq 0$, $g(\theta) = \nabla\mathcal{L}(\theta) = H(\theta - \theta^\star)$, and $e_0 = \theta^{s,0} - \theta^\star$. Stage I uses stochastic gradients $g_k = g(\theta^{s,k}) + \xi_k$ where $\mathbb{E}[\xi_k] = 0$, $\mathbb{E}\|\xi_k\|^2 \leq \sigma_f^2$, and $\mathrm{Cov}(\xi_k, \xi_\ell)$ has correlation coefficients $\rho_{|k-\ell|} \in [0, 1]$. The Stage III query at $\widetilde{\theta}^{s,K}$ has noise $\widetilde{\xi}$ with $\mathbb{E}[\widetilde{\xi}] = 0$, $\mathbb{E}\|\widetilde{\xi}\|^2 \leq \sigma_s^2$.

**Deterministic fast trajectory.** In the quadratic model, Stage I obeys $e_{k+1} = (I - \eta H)e_k$, hence $e_K = (I - \eta H)^K e_0$ and

$$\Delta_K \stackrel{\text{def}}{=} \theta^{s,K} - \theta^{s,0} = e_K - e_0 = -\left[I - (I - \eta H)^K\right] H^\dagger g_0, \qquad g_0 = g(\theta^{s,0}). \quad (14)$$

Stage II repositions to $\widetilde{\theta}^{s,K} = \theta^{s,0} + \alpha \Delta_K$. A standard Fisher/ Gauss–Newton proxy yields the drift magnitude

$$\mathrm{Drift}(K, \alpha) \propto \alpha^2 \|\Delta_K\|_H^2 = \alpha^2 \Delta_K^\top H \Delta_K = \alpha^2 g_0^\top H^\dagger \left[I - (I - \eta H)^K\right]^2 g_0. \quad (15)$$

Spectrally, for an eigenpair $(u_\lambda, \lambda > 0)$,

$$\mathrm{Drift}_\lambda(K, \alpha) = \alpha^2 \frac{\left(1 - (1 - \eta\lambda)^K\right)^2}{\lambda} \langle g_0, u_\lambda\rangle^2, \quad (16)$$

which increases monotonically with $K$ and is quadratic in $\alpha$, and saturates as $K \to \infty$.

**Stochastic components.** The fast trajectory accumulates noise $\alpha \sum_{k=0}^{K-1} \xi_k$ with

$$\mathbb{E}\left\|\alpha \sum_{k=0}^{K-1} \xi_k\right\|^2 \leq \alpha^2 \sigma_f^2 S(K, \rho), \qquad S(K, \rho) = K + 2\sum_{\Delta=1}^{K-1}(K - \Delta)\rho_\Delta \in [K, K^2], \quad (17)$$

and the slow query contributes $\mathbb{E}\|\widetilde{\xi}\|^2 \leq \sigma_s^2$.

**Informal definition.** We informally define the $\mathcal{F}(K, \alpha)$ as

$$\mathcal{F}(K, \alpha) \approx \underbrace{\alpha^2 g_0^T H^\dagger \left[ I - (I - \eta H)^K \right]^2 g_0}_{\text{off-policy drift (bias)}} + \underbrace{\alpha^2 \sigma_f^2 S(K, \rho)}_{\text{fast noise}} + \underbrace{\sigma_s^2}_{\text{slow noise}},$$

(18)

where $\approx$ means up to multiplicative constants depending only on $(H, \eta)$.

**Dependence on $K$ and $\alpha$.** For fixed $\alpha$, both drift and $S(K, \rho)$ are non-decreasing in $K$ (monotone under $0 < \eta < 1/\lambda_{\max}$), so $\mathcal{F}(K, \alpha)$ increases with $K$. For fixed $K$, $\mathcal{F}(K, \alpha)$ grows quadratically in $\alpha$. In the small-step regime ($\eta\lambda K \ll 1$), $\text{Drift}_\lambda(K, \alpha) \approx \alpha^2(\eta K)^2 \lambda \langle g_0, u_\lambda \rangle^2$, showing that excessively large $K$ or $\alpha$ makes $\mathcal{F}(K, \alpha)$ dominate the descent term.

**Implication for Stage III.** The descent-lemma bound, $\mathbb{E}[\mathcal{L}(\theta^{s+1})] \leq \mathcal{L}(\theta^{s,0}) - c\eta \|\nabla\mathcal{L}(\theta^{s,0})\|^2 + O(\eta^2 L \cdot \mathcal{F}(K, \alpha))$, treats $\mathcal{F}(K, \alpha)$ as the residual that the one-step slow update must compensate. Thus, choosing small $K$ and moderate $\alpha$ keeps $\mathcal{F}(K, \alpha)$ manageable while retaining the stability benefits of the fast trajectory.

