# OpenReview forum: "Slow-Fast Policy Optimization: Reposition-Before-Update for LLM Reasoning"
_ICLR.cc/2026/Conference — ICLR 2026 Poster_

### Official Review · Reviewer_GcTt · 2025-10-31

**Soundness:** 3
**Presentation:** 3
**Contribution:** 3
**Rating:** 6
**Confidence:** 3

**Summary:**

This paper introduces Slow-Fast Policy Optimization (SFPO), a plug-and-play method to stablize and accelerate the learning of on-policy RL algorithms such as GRPO. Specifically, SFPO has three stages for each training step: Fast Trajectory, Reposition, and Slow Correction, which mitigate the noisy single-step rollout updates and ensures a more stable optimization process. Empirical studies in mathmatical benchmarks demonstrates the consistent improvement of SFPO compared with GRPO.

**Strengths:**

1. The method is well-structured and thoughtfully designed. The Fast Trajectory stage stabilizes the optimization process, while the Reposition and Slow Correction stages help to better align the optimization with the on-policy objective. The motivation behind each stage is clearly articulated, making the overall approach sound and convincing.

2. The empirical results are thorough and well-presented. The paper evaluates SFPO across 5 models, 3 methods, and 6 benchmarks, clearly demonstrating substantial improvements in both efficacy and efficiency. Furthermore, comprehensive analyses and ablation studies are provided, strengthening the validity of SFPO’s performance claims.

3. The paper is well-organized, featuring clear and informative figures, tables, and algorithm illustrations. Additionally, Section 3 provides intuitive explanations that greatly aid in understanding the underlying mechanism of the proposed SFPO.

**Weaknesses:**

1. The empirical evaluation is somewhat limited in scope, as all experiments are conducted only on mathematical benchmarks, without including domains such as code or logic reasoning. Additionally, GRPO is the sole baseline used; incorporating additional baselines like DAPO and GSPO would further strengthen the evidence for SFPO’s effectiveness.

2. The ablation studies indicate that SFPO is sensitive to the hyperparameters $\alpha$ and $K$, which may hinder its direct applicability to domains beyond mathematical reasoning. Users may need to carefully tune these hyperparameters to achieve optimal performance.

**Questions:**

1. What does "Slow" refer to in stage 3? Does this stage require significantly more time to update the gradient?

2. Could you elaborate on the time consumption in Stage 1? Since gradient updates are computationally intensive, this stage might become slower as K increases. Additionally, in agentic scenarios involving multi-turn interactive trajectories, performing multiple policy gradient updates may significantly reduce optimization efficiency.

---

> ### Author Response · Authors · 2025-11-21
> **[1/4] Rebuttal**
>
> >**Overall**
>
> Thank you for your constructive comments. We are glad that you find our paper is "`well-structured and thoughtfully designed`", the motivation is "`is clearly articulated`", the empirical results are "`thorough and well-presented`", results analysis is "`comprehensive`", and our paper is "`well-organized, featuring clear and informative figures, tables, and algorithm illustrations`". Below, we address your questions one by one in detail. We have also **included all discussions below in our revision** (with the changed part marked in red).
>
> ---
> >**W1.1: "The empirical evaluation is somewhat limited in scope, as all experiments are conducted only on mathematical benchmarks, without including domains such as code or logic reasoning."**
> ---
> Thank you very much for raising this question regarding the generality of SFPO to other domains. We would like to humbly clarify that most reasoning RL works (e.g. GRPO, DAPO [1, 2] ) primarily focus on mathematical reasoning benchmarks, and we follow this established setting to ensure a fair and direct comparison with existing methods.
>
> However, we acknowledge that restricting the evaluation to math benchmarks does not fully demonstrate the generality of SFPO. Following your suggestion, We additionally trained DeepSeek-R1-Distill-Qwen-1.5B on the Skywork-OR1 Code RL training dataset [3] with 14.1k data. The training hyperparameters mirror those used for math tasks in submitted manuscript, with $k=3$ and $\alpha=0.8$ by default. We then evaluate the resulting checkpoints on LiveCodeBench [4] with a 32K response length and report Pass@1 (Avg 4). The results are summarized in the table below.
>
> |      | 0     | 50    | 100   | 150   | 200   | 250   | 300   | 350   | 400   |
> |------|-------|-------|-------|-------|-------|-------|-------|-------|-------|
> | GRPO | 17.02 |16.48 | 17.18 | 17.27 | 17.38 | 17.47 | 17.56 | 16.31 | 18.91 |
> | SFPO | 17.02 |**16.49**| **17.83** | **17.92** | **19.40** | **19.08** | **19.62** | **21.77** | **20.52** |
>
> As shown, SFPO consistently outperform GRPO throughout the whole training process. GRPO reaches a maximum Pass@1 (Avg 4) of 18.91 at 400 steps, whereas SFPO attains a higher peak performance of 21.77 at 350 steps. This consistent improvement on the coding benchmark provides additional evidence that the efficacy of SFPO extends beyond purely mathematical reasoning tasks.
>
> We also conduct ablation study to investigate the impact of $\alpha$ and $K$ on coding tasks. Similar to math tasks, we consider two scenarios: **(i)** a small inner-update budget $K = 3$ with varying $\alpha \in {0.2, 0.8, 1.0}$, and **(ii)** a larger budget $K = 7$ with the same set of $\alpha$ values. As shown in the table below, when $K=3$, the default value $\alpha=0.8$ achieves the best accuracy through out the training process, which is consistent with the findings in math tasks. While for $K=7$, a smaller $\alpha=0.2$ is preferred to mitigate stronger distribution drift and restore stability, again aligning with the trends observed in math RL training.
>
> | Setting                     |   0   |   50  |   100 |   150 |   200 |
> |----------------------------|------:|------:|------:|------:|------:|
> | GRPO                       | 17.02 | 16.48 | 17.18 | 17.27 | 17.38 |
> | SFPO (k = 3, α = 0.2)      | 17.02 | 17.47 | 17.47 | 18.10 | 18.28 |
> | SFPO (k = 3, α = 0.8)      | 17.02 | 16.49 | 17.83 | 17.92 | 19.40 |
> | SFPO (k = 3, α = 1.0)      | 17.02 | 17.56 | 18.54 | 17.20 | 16.95 |
>
> | Setting                     |   0   |   50  |   100 |   150 |   200 |
> |----------------------------|------:|------:|------:|------:|------:|
> | GRPO                       | 17.02 | 16.48 | 17.18 | 17.27 | 17.38 |
> | SFPO (k = 7, α = 0.2)      | 17.02 | 18.01 | 19.44 | 22.49 | 22.13 |
> | SFPO (k = 7, α = 0.8)      | 17.02 | 17.02 | 19.53 | 20.43 | 19.35 |
> | SFPO (k = 7, α = 1.0)      | 17.02 | 16.75 | 16.67 | 18.01 | 18.27 |
>
> The above discussions and results have also been added to Appendix C.2 in the updated manuscript.

---

> ### Author Response · Authors · 2025-11-21
> **[2/4] Rebuttal**
>
> ---
> >**W1.2 Additionally, GRPO is the sole baseline used; incorporating additional baselines like DAPO and GSPO would further strengthen the evidence for SFPO’s effectiveness.**
> ---
>
> Thank you for your suggestions. We conduct additional experiments on DAPO baselines, comparing DAPO+SFPO with vanilla DAPO using DeepSeek-R1-Distill-Qwen-1.5B and Qwen3-4B-Base as the policy models.The validation accuracy on the math benchmarks over the course of training is shown in Appendix Figure 11. From these curves, we observe that for both models, **SFPO not only consistently attains higher accuracy than DAPO throughout training, but also reaches the target performance substantially earlier**: roughly 225 steps sooner on DS-R1-Distill-Qwen-1.5B and 120 steps sooner on Qwen3-4B-Base.
>
> We further summarize the best accuracy achieved during training in the Table below, where for each method we select the checkpoint with the highest average score across all math benchmarks. On DS-R1-Distill-Qwen-1.5B, SFPO attains an average accuracy of 50.56, while DAPO reaches 49.30; on Qwen3-4B-Base, SFPO achieves 47.87 compared to 46.48 for DAPO, with consistent improvements across most individual benchmarks.
>
> | Model                      | Method | Math-500 | AIME24 | AIME25 |  AMC  | Minerva | Olympiad |  Avg  |
> |---------------------------|--------|---------:|-------:|-------:|------:|--------:|---------:|------:|
> | **DS-R1-distilled-Qwen-1.5B** | DAPO   |  85.40  | 32.50  | 24.17  | 68.07 |  33.64  |  52.00   | 49.30 |
> |                           | SFPO   |  **86.25**  | **33.33**  | **26.67**  | **70.18** |  **34.38**  |  **52.52**   | **50.56** |
> | **Qwen3-4B-Base**         | DAPO   |  84.50  | 22.50  | 20.83  | **59.64** |  39.98  |  51.41   | 46.48 |
> |                           | SFPO   |  **85.55**  | **25.83**  | **23.33**  | 58.13 |  **41.27**  |  **53.12**   | **47.87** |
>
> We have added the above comparison results to the Appendix C.4 in the revision as suggested.
>
> ---
> >**W2: "The ablation studies indicate that SFPO is sensitive to the hyperparameters K and \alpha, which may hinder its direct applicability to domains beyond mathematical reasoning. Users may need to carefully tune these hyperparameters to achieve optimal performance."**
> ---
>
> We appreciate the reviewer’s concern regarding the dependence of SFPO on the hyperparameters $K$ and $\alpha$. We would like to clarify that SFPO does not require aggressive parameter tuning in practice.
>
> While SFPO does introduce two additional hyperparameters, our ablation studies across both mathematical and coding tasks (as shown in reply to Q1.1) show that SFPO is robust within a small and intuitive region of the $[K,\alpha]$ space. In particular, we find that:
>
> (1) A small value of $K$ (e.g., $K=3$) already captures most of the benefit of the slow–fast lookahead, and
>
> (2) A moderately large $\alpha$ (e.g., $\alpha=0.8$) consistently works well.
>
> Our default setting $K=3, \alpha=0.8$ was chosen precisely because it performs strongly across all mathematical and coding tasks. Larger $K$ does not bring additional gains but only increases compute, which suggests that SFPO is more effective in the small $K$ regime, rather than requiring delicate fine-tuning.
>
> ---
> >**Q1: "What does "Slow" refer to in stage 3? Does this stage require significantly more time to update the gradient?"**
> ---
> We apologize for the confusion. The terms ``fast`` in stage 1 and ``slow`` in Stage 3 are simply naming choices.
>
> In our terminology, *slow* refers to the frequency of updates, not the computational speed. Specifically, Stage 1 performs multiple fast, high-frequency updates (K inner steps), whereas Stage 3 performs only a single correction update. Thus, Stage 1 is called *fast* due to its update frequency, while Stage 3 is called *slow* because it occurs only once per iteration.
>
> Since slow refers solely to the update frequency, it does not imply any additional computational overhead. Stage 3 is simply a standard one-step GRPO update, and therefore does not require more time than a normal GRPO iteration.
>
> We have added this clarification to the Appendix D.2 in the revision.

---

> ### Author Response · Authors · 2025-11-21
> **[3/4] Rebuttal**
>
> ---
> >**Q2.1: "Could you elaborate on the time consumption in Stage 1? Since gradient updates are computationally intensive, this stage might become slower as K increases."**
> ---
>
> We thank for the question about the time consumption of Stage I. We provide a more comprehensive analysis below:
>
> **Does per-iteration compute increase with $K$?**
>
> Yes. In SFPO, each iteration performs $K$ fast policy updates on the same batch, so the backward-pass compute for the policy update scales approximately linearly with $K$. Previous works have shown that under GRPO, about **70%** of the per-step time is spent on rollout generation, **≈20%** on the policy update, and **≈10%** on other overhead. Increasing $K$ from 1 to 3 multiplies only the 20% “policy-update” portion, so the worst-case theoretical overhead is
>
> $$
> \frac{T_{\text{SFPO}}}{T_{\text{GRPO}}}
> \approx 1 + (K-1)\cdot 0.2
> = 1.4.
> $$
>
> In practice, measured average per-step wall-clock times on DeepSeek-R1-Distill-Qwen-1.5B are $\sim300$ s ($\sim240$ s for rollout generation, $\sim40$ s for actor update) for GRPO and $\sim410$ s ($\sim240$ s for rollout generation, $\sim150$ s for actor update) for SFPO (with $K=3,\alpha=0.8$), i.e., SFPO steps are only **≈1.21×** more expensive than GRPO steps. Moreover, our ablations on $\alpha$ and $K$ over both math and coding benchmarks show that this relatively small $K=3$ with a moderately large $\alpha=0.8$ is already sufficient; larger $K$ does **not** yield further gains and is therefore unnecessary in practice.
>
>
> **Matched wall-clock: fixed time budget.**
>
> To compare at matched compute, we convert training steps to elapsed wall-clock time and plot validation accuracy versus time as shown in in the Table below. On  DeepSeek-R1-Distill-Qwen-1.5B, for any reasonable time budget between 5 and 33 hours, SFPO consistently outperforms GRPO:
>
> | Time (h) | GRPO acc | SFPO acc | acc gain |
> |---------:|:--------:|:--------:|:-----------:|
> | 5        | 43.75   | 46.23   | **+2.48** |
> | 10       | 45.76   | 46.84   | **+1.08** |
> | 15       | 44.81   | 48.41   | **+3.60** |
> | 20       | 45.79   | 47.64   | **+1.85** |
> | 25       | 45.14   | 47.75   | **+2.61** |
> | 30       | 46.20   | 48.17   | **+1.97** |
> | 33       | 46.71   | 47.44   | **+0.74** |
>
> That is, **at the same wall-clock time** SFPO is typically **1–3.6 points** more accurate than GRPO, despite its relatively higher per-step compute.
>
> **Matched wall-clock: fixed accuracy targets.**
>
> We would first like to clarify that “GRPO’s best accuracy” in our original experiments was **already used as a fixed accuracy target**: we took the maximum validation accuracy achieved by GRPO during training and then measured how much wall-clock time SFPO needs to reach this same target.
>
> To make the comparison more flexible, we now also consider multiple fixed accuracy targets and ask how long each method needs to reach a given accuracy. On  DeepSeek-R1-Distill-Qwen-1.5B we obtain:
>
> | Target acc | $t_{\text{GRPO}}$ (h) | $t_{\text{SFPO}}$ (h) | Speedup (GRPO / SFPO) |
> |-----------:|:-----------------------:|:------------------------:|:----------------------:|
> | 0.44       | 5.78                    | 1.81                     | **3.19×**              |
> | 0.45       | 7.96                    | 2.21                     | **3.60×**              |
> | 0.46       | 29.55                   | 4.50                   | **6.57×**              |
> | 0.47       | 32.40                   | 10.56                  | **3.07×**              |
>
> Thus, to reach the same accuracy, SFPO requires **3–6.6× less wall-clock time**.
>
> In summary, while Stage I of SFPO does incur additional gradient updates and thus increases per-iteration compute as (K) grows, we deliberately operate in a regime with a small (K) where the per-step overhead is modest. Under fair comparisons at matched wall-clock time and at fixed accuracy targets, SFPO is substantially more compute-efficient than GRPO, indicating that the extra Stage-I cost is more than compensated by improved sample and optimization efficiency.
>
> We have added the above analysis to the Appendix C.5 in the revision.

---

> ### Author Response · Authors · 2025-11-21
> **[4/4] Rebuttal**
>
> ---
> >**Q2.2: "Additionally, in agentic scenarios involving multi-turn interactive trajectories, performing multiple policy gradient updates may significantly reduce optimization efficiency."**
> ---
>
> Thank you very much for this interesting thought. Our work focuses on single-turn, RLHF-style reasoning tasks (math and coding), and we do not explicitly consider fully agentic, multi-turn interactive scenarios in this paper, as they are **beyond our current scope**. Conceptually, SFPO is orthogonal to whether GRPO is used in a single-turn or multi-turn formulation: SFPO modifies the update rule (slow–fast updates plus interpolation), and it does not assume a particular reward structure. In principle, one could plug a multi-turn GRPO objective directly into Stage I of SFPO and obtain a multi-turn variant. However, such a variant would inherit the same limitation of using only trajectory-level rewards, unless one additionally designs a better multi-turn advantage estimator. [1,2]
>
> Regarding efficiency, multiple fast updates do increase the policy-update cost, but this overhead is moderate for small $K$ as analyzed in the previous question. In long-horizon, agentic settings one could also choose a smaller $K$, so that SFPO remains very close to GRPO in compute while still providing efficiency benefits.
>
> In summary, a thorough empirical study of SFPO in fully agentic, multi-turn environments, together with more expressive advantage estimators that leverage intermediate rewards, is an important and interesting direction for future work, which we would like to leave this for future research.
>
> [1] Wei, Q., Zeng, S., Li, C., Brown, W., Frunza, O., Deng, W., Schneider, A., Nevmyvaka, Y., Zhao, Y. K., Garcia, A., & Hong, M. (2025). Reinforcing Multi-Turn Reasoning in LLM Agents via Turn-Level Reward Design. arXiv preprint arXiv:2505.11821.
>
> [2] Wang, R., & Ammanabrolu, P. (2025). A Practitioner’s Guide to Multi-turn Agentic Reinforcement Learning. arXiv preprint arXiv:2510.01132.

---

### Official Review · Reviewer_5eGS · 2025-10-31

**Soundness:** 3
**Presentation:** 3
**Contribution:** 2
**Rating:** 4
**Confidence:** 5

**Summary:**

This paper introduces Slow-Fast Policy Optimization (SFPO), a three-stage update mechanism designed to improve the stability and sample efficiency of on-policy reinforcement learning for LLM reasoning. SFPO replaces the standard single gradient step with a fast trajectory of multiple inner updates, a reposition step to control off-policy drift, and a final slow correction step. Extensive experiments demonstrate that SFPO significantly outperforms the GRPO baseline in accuracy, convergence speed, and wall-clock time on various math reasoning benchmarks.

**Strengths:**

1.  The proposed method addresses a significant and well-recognized problem: the instability and sample inefficiency of on-policy algorithms like GRPO in the context of LLM reasoning. The proposed three-stage solution is simple, intuitive, and clearly motivated as a way to reduce update variance and make better use of collected data.
2.  A major strength of SFPO is its "plug-and-play" compatibility with existing policy gradient pipelines. By modifying only the parameter update rule while leaving the loss function, rollout generation, and regularization terms of GRPO intact, the method offers a practical, low-effort path to improving existing training setups.
3.  The paper is supported by a comprehensive set of experiments that demonstrate consistent and substantial gains over a strong GRPO baseline across multiple models and benchmarks. The reported improvements in sample efficiency (up to $4.93\times$ fewer rollouts) and wall-clock time (up to $4.19\times $ reduction) are highly significant. Furthermore, the ablation studies effectively justify the inclusion of each component of SFPO, particularly the adaptive entropy control and the slow correction stage.

**Weaknesses:**

1.  The reporting of experimental results contains significant numerical errors and omissions. In Section 4.2.1, the claimed performance gains for Qwen2.5-Math-7B (+1.80) and DS-distilled-Qwen-7B (+0.8) do not match the values in Table 1, which show gains of +0.83 and +2.57, respectively.
2.  The reference list is not properly curated. For example, the citations "Yu et al., 2025a" and "Yu et al., 2025b" both point to the exact same arXiv preprint, which is misleading.

**Questions:**

1.  Could you clarify the nomenclature for Stage I, the 'Fast Trajectory'? The term 'fast' is somewhat ambiguous. Does it refer to the speed of convergence, the magnitude of the parameter change, or some other property?
2.  A crucial baseline seems to be missing from the ablation studies. To isolate the benefits of Stage II (Reposition) and Stage III (Slow Correction), have you considered a baseline that only uses Stage I? This would be equivalent to applying GRPO with $K$ inner updates on the same batch, which would more clearly demonstrate the contribution of the reposition and slow correction steps.
3.  Stage II is designed to control off-policy drift via interpolation. A more common approach in policy gradient methods is to use a KL divergence penalty to constrain the policy update. Could you comment on the design choice of interpolation over a KL penalty? Have you compared the performance of your reposition step against a KL-constrained update in this multi-step update setting?
4.  The paper claims that SFPO addresses the high-variance gradient issue in GRPO. A significant source of variance in GRPO can come from batches with uniform rewards (e.g., all successes or all failures), which result in zero advantage for all samples. Does SFPO's performance gain stem primarily from better handling these 'zero-advantage' batches? To verify this, it might be necessary to conduct experiments with an algorithm like DAPO, which is explicitly designed to use *dynamic sampling* to mitigate this issue, to see if SFPO provides similar benefits in a setting where this specific source of variance is already addressed.
5.  Could you provide a derivation or at least a reference for the descent guarantee in Equation (10)? A more formal explanation of the assumptions and the term $F(K, \alpha) $ would strengthen the paper's theoretical grounding.
6.  Would you be able to correct the numerical errors in the text describing the results in Table 1 (lines 322-323)? The discrepancies between the reported gains and the values in the table are substantial.
7. The interpretation of lower entropy in SFPO as "more efficient exploration" (lines 374-375) is intriguing. Could you elaborate on this? Is it possible that the method simply encourages faster convergence to a more deterministic policy, which might be beneficial for exploitation but could also be interpreted as a sign of reduced exploration?

I look forward to the author's response. I am willing to reconsider my score if the questions above, particularly those concerning the experimental details, numerical correctness, and key design choices, are adequately addressed.

---

> ### Author Response · Authors · 2025-11-21
> **[1/4] Rebuttal**
>
> >**Overall**
>
> Thank you for your constructive comments. We are glad that you find our problem "`significant and well-recognized`", our method "`simple, intuitive, and clearly motivated`", our experiments are "`comprehensive`", performance is "`highly significant`", and ablation studies are "`effectively`" justify our method. Below, we address your questions one by one in detail. We have also **included all discussions below in our revision** (with the changed part marked in red).
>
> ---
> >**W1: "The reporting of experimental results contains significant numerical errors and omissions. In Section 4.2.1, the claimed performance gains for Qwen2.5-Math-7B (+1.80) and DS-distilled-Qwen-7B (+0.8) do not match the values in Table 1, which show gains of +0.83 and +2.57, respectively."**
>
> >**Q6:"Would you be able to correct the numerical errors in the text describing the results in Table 1 (lines 322-323)? The discrepancies between the reported gains and the values in the table are substantial."**
> ---
>
> Thank you very much for carefully checking our results and pointing out these numerical inconsistencies. We have corrected these numbers in the revised manuscript so that the text is fully consistent with Table 1 (lines 322–323 and Table 1).
>
> ---
> >**W2: "The reference list is not properly curated. For example, the citations "Yu et al., 2025a" and "Yu et al., 2025b" both point to the exact same arXiv preprint, which is misleading."**
> ---
> We apologize for the reference issue, and we have corrected it in the revised manuscript.
>
> ---
> >**Q1: "Could you clarify the nomenclature for Stage I, the 'Fast Trajectory'? The term 'fast' is somewhat ambiguous. Does it refer to the speed of convergence, the magnitude of the parameter change, or some other property?"**
> ---
> We apologize for the confusion. The terms ``fast`` in stage 1 and ``slow`` in Stage 3 are simply naming choices.
>
> In our terminology, *slow* refers to the frequency of updates, not the computational speed. Specifically, Stage 1 performs multiple fast, high-frequency updates (K inner steps), whereas Stage 3 performs only a single correction update. Thus, Stage 1 is called *fast* due to its update frequency, while Stage 3 is called *slow* because it occurs only once per iteration.
>
> Since fast refers solely to the update frequency, it **does not imply any other property**.
>
> We have added this clarification to the Appendix D.2 in the revision.
>
> ---
> >**Q2: "A crucial baseline seems to be missing from the ablation studies. To isolate the benefits of Stage II (Reposition) and Stage III (Slow Correction), have you considered a baseline that only uses Stage I? This would be equivalent to applying GRPO with K inner updates on the same batch, which would more clearly demonstrate the contribution of the reposition and slow correction steps."**
> ---
>
> Thank you for pointing out this important ablation study. We would like to humbly clarify that we have already included this ablation as shown in Figure 5(a) and Figure 5(b) of manuscript. In both figures, $\alpha=1.0$ corresponds to removing the reposition step (see Eq. 7 in line 193 in the main paper for details). As shown there, SFPO performs significantly worse without Stage II, confirming the importance of the interpolation step.
>
> To make the significance of Stage II much clearer, we further conduct an explicit comparison between SFPO with and without the interpolation scheme using DeepSeek-R1-Distill-Qwen-1.5B, and the results are demonstrated in the two Tables below.
>
> - When $K=3$, SFPO without Stage II is slightly better at the very beginning of training because the drift accumulated over only three fast updates remains relatively small. However, as training progresses, its performance begins to degrade due to the growing off-policy mismatch. Since $K=3$ is still relatively close to the vanilla GRPO, the degradation in the later stage becomes comparable to standard GRPO.
> - When $K=7$, The difference becomes substantially more pronounced. SFPO without Stage II initially gains some accuracy, but the lack of a reposition step causes severe off-policy drift as training continues. With $K=7$, this drift grows large enough that the method moves far away from the on-policy region used by standard GRPO, eventually causing the training to collapse. In contrast, SFPO with Stage II remains stable throughout training and consistently achieves the best overall performance.
>
> These results confirm that the interpolation-based reposition step in Stage II is crucial for stabilizing training process and turning the additional inner updates into consistent gains. The above discussion and results have also added to the Appendix C.1 in the updated manuscript.

---

> ### Author Response · Authors · 2025-11-21
> **[2/4] Rebuttal**
>
> | Method                     |   0    |   20    |   40    |   60    |   80    |   100   |   120   |   140   |   160   |   180   |   200   |   220   |   240   |   260   |   280   |   300   |   320   |   340   |   360   |   380   |   400   |
> |---------------------------|--------|---------|---------|---------|---------|---------|---------|---------|---------|---------|---------|---------|---------|---------|---------|---------|---------|---------|---------|---------|---------|
> | **GRPO**                  | 0.3727 | 0.3975  | 0.4387  | 0.4374  | 0.4442  | 0.4531  | 0.4584  | 0.4497  | 0.4546  | 0.4462  | 0.4533  | 0.4542  | 0.4594  | 0.4523  | 0.4551  | 0.4492  | 0.4458  | 0.4578  | 0.4603  | 0.4653  | 0.4602  |
> | **SFPO w/o Stage II (K=3)**| 0.3727 | **0.4492**  | **0.4640**  | **0.4681**  | **0.4810**  | **0.4850**  | **0.4996**  | 0.4839  | 0.4895  | 0.4890  | 0.4787  | 0.4831  | 0.4748  | 0.4737  | 0.4734  | 0.47654  | 0.4641  | 0.4650  | 0.4632  | 0.4664  | 0.4694  |
> | **SFPO w/ Stage II (K=3)**| 0.3727| 0.4344  | 0.4563  | 0.4564  | 0.4654  | 0.4713  | 0.4821  | **0.4857**  | **0.4938**  | **0.4901**  | **0.4885**  | **0.4928**  | **0.4934**  | **0.4822**  | **0.5025**  | **0.5014**  | **0.5082**  | **0.4900**  | **0.4978**  | **0.4953**  | **0.5063**  |
>
>
> | Method                         |   0    |   20    |   40    |   60    |   80    |   100   |   120   |   140   |   160   |   180   |   200   |
> |--------------------------------|--------|---------|---------|---------|---------|---------|---------|---------|---------|---------|---------|
> | **GRPO**                       | 0.3727 | 0.4387  | 0.4442  | 0.4584  | 0.4546  | 0.4693  | 0.4594  | 0.4551  | 0.4568  | 0.4653  | 0.4602  |
> | **SFPO w/o Stage II (K=7)**    | 0.3727 | **0.4808**  | **0.4795**  | **0.4743**  | 0.4357  | 0.4265  | 0.4078  | 0.3974  | 0.4022  | 0.4137  | 0.4022  |
> | **SFPO w/ Stage II (K=7)**     | 0.3727 | 0.4246  | 0.4489  | 0.4656  | **0.4659**  | **0.4754**  | **0.4988**  | **0.4795**  | **0.4891**  | **0.4947**  | **0.4967**  |
>
> ---
> >**Q3: "Stage II is designed to control off-policy drift via interpolation. A more common approach in policy gradient methods is to use a KL divergence penalty to constrain the policy update. Could you comment on the design choice of interpolation over a KL penalty? Have you compared the performance of your reposition step against a KL-constrained update in this multi-step update setting?"**
> ---
>
> Thank you very much for this thoughtful comment regarding the effectiveness of interpolation in stage II against KL divergence penalty.
>
> **Why Stage 2 (Reposition) is necessary beyond KL penalty?**
>
> In Stage 1, SFPO performs $K$ consecutive updates using the same rollout batch. This means that the parameter $\theta^{s,K}$ can drift far from the rollout-generating point $\theta^{s,0}$, creating a strong off-policy mismatch. A KL penalty cannot reliably prevent this issue: KL regularizes each step locally, but when $K$ grows, the cumulative drift may still become very large, and $\theta^{s,K}$ eventually moves far away from $\theta^{s,0}$ regardless of the KL strength. Stage II directly addresses this problem by **extracting only the direction** of the fast trajectory while discarding the off-policy magnitude. Specifically, we reposition the model back toward $\theta^{s,0}$, which ensures that the updated point remains sufficiently close to $\theta^{s,0}$ so that the original rollout is still on-policy enough for the next GRPO update (stage III). This “direction-only” mechanism cannot be achieved by a KL penalty, which cannot reset the accumulated drift. Therefore, Stage II is a natural solution to off-policy drift, whereas KL penalty provides only local regularization and cannot prevent large cumulative mismatch when reusing rollouts multiple times.
>
> We further conducted additional experiments comparing the KL-penalty with the interpolation scheme to support the above explanation. Specifically, we use Qwen3-4B-Base as the policy model, and the training dataset and all other settings are identical to those in Section 4.1 of the submitted manuscript. We then compare interpolation against the KL-penalty over mathematical reasoning benchmarks under two inner update settings, with $K=3, \alpha=0.8$ and $K=7, \alpha=0.2$, respectively.
>
> The results are shown in the Table below. We can observe that for k=3, the interpolation consistently outperforms the KL penalty variant throughout the training; for example, at step 200 the interpolation reaches **45.44**, compared to **40.64** with the KL penalty. The gap becomes even more pronounced when k=7: the interpolation scheme remains stable and continues to improve up to **46.14** at step 200, whereas the KL-penalty variant becomes unstable and collapses after step 80, yielding zero scores. These results empirically demonstrate our interpolation scheme in Stage II provides both better performance and substantially improved stability compared to a KL-constrained update.

---

> ### Author Response · Authors · 2025-11-21
> **[3/4] Rebuttal**
>
> | Method  | 0     | 20    | 40   | 60   | 80   | 100   | 120   | 140   | 160   |180   |200   |
> |------|-------|-------|-------|-------|-------|-------|-------|-------|-------|-------|-------|
> | GRPO | 17.81 |36.47 | 37.93 | 38.74 | 38.68 | 40.01 | 39.62 | 41.43 | 40.94 |40.76 |42.10 |
> | w/ interpolation (K=3) | 17.81 | **37.71** | **40.06** | **44.27** | **44.66** | **45.58** | **44.77** | **43.90** | **45.54** |**44.97** |**45.44**|
> | w/ KL Penalty (K=3) | 17.81 |35.61 | 37.60 | 40.60 | 39.24 | 37.02 | 37.27 | 38.51 | 39.01 |40.28 |40.64 |
>
> | Method  | 0     | 20    | 40   | 60   | 80   | 100   | 120   | 140   | 160   |180   |200   |
> |------|-------|-------|-------|-------|-------|-------|-------|-------|-------|-------|-------|
> | GRPO | 17.81 |**36.47** | 37.93 | 38.74 | 38.68 | 40.01 | 39.62 | 41.43 | 40.94 |40.76 |42.10 |
> | w/ interpolation (K=7) | 17.81 | 31.30 | **40.44** | **43.91** | **40.67** | **41.78** | **42.68** | **44.42** | **43.96** |**44.16** |**46.14**|
> | w/ KL Penalty (K=7) | 17.81 |33.30 | 35.19 | 36.34 | 00.00 | 00.00 | 00.00 | 00.00 | 00.00 |00.00 |00.00 |
>
> We have also included these results in Appendix C.3 in the revision.
>
> ---
> >**Q4: "The paper claims that SFPO addresses the high-variance gradient issue in GRPO. A significant source of variance in GRPO can come from batches with uniform rewards (e.g., all successes or all failures), which result in zero advantage for all samples. Does SFPO's performance gain stem primarily from better handling these 'zero-advantage' batches? To verify this, it might be necessary to conduct experiments with an algorithm like DAPO, which is explicitly designed to use dynamic sampling to mitigate this issue, to see if SFPO provides similar benefits in a setting where this specific source of variance is already addressed."**
> ---
>
> Thank you very much for these insightful comments. We would like to clarify that SFPO’s performance gains do **not** primarily stem from better handling of “zero-advantage” batches. Instead, SFPO operates at a **higher-level optimization layer** that is independent of the internal mechanics of GRPO or DAPO; it is therefore **orthogonal** to these methods and can be seamlessly integrated with GRPO, DAPO, or any GRPO-family variant.
>
> **Why SFPO achieves better performance?**
>
> SFPO’s improved efficacy over vanilla GRPO relies on three key design components:
>
> (1) Stage I produces a more stable and accurate update direction by reusing the same rollout batch for (K) fast updates. Although $\theta^{s,K}$ may move far from the rollout-generating point $\theta^{s,0}$.
>
> (2) Stage II then **extracts only the direction** of this fast trajectory and repositions the model back toward $\theta^{s,0}$ using a small, controllable step size $\alpha$. This keeps the update sufficiently on-policy while still leveraging the stabilized directional information.
>
> (3) After repositioning, Stage III performs a standard GRPO update, effectively guiding each iteration toward a better-conditioned and more accurate point before the actual policy update is applied. Repeating this procedure makes early rollouts stronger and accelerates convergence.
>
> To provide empirical support for the above explanation, we conduct additional experiments on DAPO baselines, comparing DAPO+SFPO with vanilla DAPO using DeepSeek-R1-Distill-Qwen-1.5B and Qwen3-4B-Base as the policy models. The validation accuracy on the math benchmarks over the course of training is shown in Appendix Figure 12. From these curves, we observe that for both models, **SFPO not only consistently attains higher accuracy than DAPO throughout training, but also reaches the target performance substantially earlier:** roughly 225 steps sooner on DS-R1-Distill-Qwen-1.5B and 120 steps sooner on Qwen3-4B-Base. We further summarize the best accuracy achieved during training in the Table below, where for each method we select the checkpoint with the highest average score across all math benchmarks. On DS-R1-Distill-Qwen-1.5B, SFPO attains an average accuracy of 50.56, while DAPO reaches 49.30; on Qwen3-4B-Base, SFPO achieves 47.87 compared to 46.48 for DAPO, with consistent improvements across most individual benchmarks.
>
> | Model                      | Method | Math-500 | AIME24 | AIME25 |  AMC  | Minerva | Olympiad |  Avg  |
> |---------------------------|--------|---------:|-------:|-------:|------:|--------:|---------:|------:|
> | **DS-R1-distilled-Qwen-1.5B** | DAPO   |  85.40  | 32.50  | 24.17  | 68.07 |  33.64  |  52.00   | 49.30 |
> |                           | SFPO   |  **86.25**  | **33.33**  | **26.67**  | **70.18** |  **34.38**  |  **52.52**   | **50.56** |
> | **Qwen3-4B-Base**         | DAPO   |  84.50  | 22.50  | 20.83  | **59.64** |  39.98  |  51.41   | 46.48 |
> |                           | SFPO   |  **85.55**  | **25.83**  | **23.33**  | 58.13 |  **41.27**  |  **53.12**   | **47.87** |

---

> ### Author Response · Authors · 2025-11-21
> **[4/4] Rebuttal**
>
> Since DAPO already employs dynamic sampling to mitigate the zero-advantage issue, these results indicate that SFPO continues to provide clear benefits even in a setting where this particular source of variance is explicitly addressed. This suggests that SFPO’s gains do not primarily stem from handling zero-advantage batches. We have also added this discussion and results to the revised manuscript in Appendix C.4.
>
> ---
> >**Q5: "Could you provide a derivation or at least a reference for the descent guarantee in Equation (10)? A more formal explanation of the assumptions and the term F(K, \alpha) would strengthen the paper's theoretical grounding."**
> ---
> Thank you for your question. Please note that Eq. (10) is intended as theoretical intuition for Stage 3 rather than a formal theorem, and a complete proof is beyond the scope of this work.
>
> To improve clarity, we have revised the section title from ``theoretical intuition`` to ``intuition``, and we now provide additional informal insights to Appendix D.4 in the revised paper. A brief summary is given below:
>
> **Informal definition** We *informally* defined the $\mathcal{F}(K,\alpha)$ as
>
> $\mathcal{F}(K, \alpha) \approx \alpha^2 g_{0}^{T} H^{\dagger} \Big[ I - ( I- \eta H)^K \Big]^2 g_0 + \alpha^2 \sigma_f^2 S(K,\rho) + \sigma_s^2$,
>
> where $\approx$ means up to multiplicative constants depending only on $(H,\eta)$. Here, for each component,
>
> - $\alpha^2 g_{0}^{T} H^{\dagger} \Big[ I - ( I- \eta H)^K \Big]^2 g_0$ is the off-policy drift (bias),
> - $\alpha^2 \sigma_f^2 S(K,\rho)$ is the fast noise, and
> - $\sigma_s^2$ is the slow noise.
>
> **Dependence on $K$ and $\alpha$** For fixed $\alpha$, both drift and $S(K,\rho)$ are non-decreasing in $K$ (monotone under $0 < \eta < 1/\lambda_{\max}$), so $\mathcal{F}(K,\alpha)$ increases with $K$.
> For fixed $K$, $\mathcal{F}(K,\alpha)$ grows quadratically in $\alpha$.
> In the small-step regime ($\eta \lambda \ll 1$),
> $\mathrm{Drift} (K,\alpha) \approx \alpha^2 (\eta K)^2 \lambda \langle g_0, u_{\lambda} \rangle^2$,
> showing that excessively large $K$ or $\alpha$ makes $\mathcal{F}(K,\alpha)$ dominate the descent term.
>
> **Implication for Stage~III.** The descent-lemma bound,
> $\mathbb{E} [\mathcal{L}(\theta^{s+1})] \le \mathcal{L}(\theta^{s,0}) - c \eta \| \nabla \mathcal{L}(\theta^{s,0}) \|^{2} + O(\eta^2 L \cdot \mathcal{F}(K,\alpha))$, treats $\mathcal{F}(K,\alpha)$ as the residual that the one-step slow update must compensate.
> Thus, choosing **small $K$** and **moderate $\alpha$** keeps $\mathcal{F}(K,\alpha)$ manageable while retaining the stability benefits of the fast trajectory.
>
> ---
> >**Q7: "The interpretation of lower entropy in SFPO as "more efficient exploration" (lines 374-375) is intriguing. Could you elaborate on this? Is it possible that the method simply encourages faster convergence to a more deterministic policy, which might be beneficial for exploitation but could also be interpreted as a sign of reduced exploration?"**
> ---
> Thank you for this thoughtful question. We agree that, in general, lower entropy can sometimes indicate premature convergence to an over-deterministic policy. In our setting, however, the observed entropy reduction in SFPO is better interpreted as more efficient exploration, rather than collapsing exploration.
>
> In vanilla GRPO, each update depends on a single-step, high-variance advantage estimate, which makes the early update direction unstable. As a result, GRPO often keeps higher entropy to compensate for this noise.
>
> In contrast, SFPO aggregates multiple gradient steps (Stage I), producing a much more stable and accurate update direction. Given this reduced variance, the policy can more confidently up-weight promising actions and down-weight clearly suboptimal ones, so it does not need to maintain as much randomness to keep improving. Importantly, SFPO still retains GRPO’s entropy bonus, so it does not suppress exploration intentionally.
>
> Overall, the lower entropy we observe corresponds to more efficient exploration, not reduced exploration. The performance improvements and stable diversity metrics strongly support this interpretation.
>
> We have added the above discussion to the Appendix D.3 in the revision.

---

> > ### Comment · Reviewer_5eGS · 2025-11-26
> >
> > The additional experiments provided in the rebuttal sufficiently address my original concern. I have now updated my recommendation from borderline reject to borderline accept.

---

> > > ### Author Response · Authors · 2025-11-26
> > >
> > > Dear Reviewer 5eGS,
> > >
> > > Thank you very much for your careful reevaluation after the rebuttal. We truly appreciate your engagement with our work and are glad that our clarifications helped address your original concerns.
> > >
> > > If there are any remaining issues that still keep the paper at a borderline accept rather than a clear accept in your view, we would be very grateful for your guidance. We are happy to address any remaining concerns you may have.
> > >
> > > Thank you again for your valuable time and constructive feedback.
> > >
> > > Best regards,
> > >
> > > Authors of 14321

---

### Official Review · Reviewer_JUEZ · 2025-11-03

**Soundness:** 3
**Presentation:** 3
**Contribution:** 3
**Rating:** 6
**Confidence:** 4

**Summary:**

This paper argues that the direction of token-level updates, measured by the signed log-probability difference between base and RLVR models, is more informative than magnitude-based metrics for understanding RLVR's effect on LLM reasoning. The authors propose and validate two methods, test-time extrapolation and training-time advantage reweighting, that exploit the directional insight to improve reasoning performance.

**Strengths:**

The paper introduces a novel and intuitive directional metric that effectively captures sparse, reasoning-critical updates, supported by rigorous token-replacement experiments and gradient analysis.

 The proposed methods are simple yet effective, demonstrating consistent gains across multiple models and benchmarks without requiring additional training data.

**Weaknesses:**

The test-time extrapolation method requires access to both the base and RLVR models, which may limit its practicality in settings where only the fine-tuned model is available.

    The paper focuses primarily on mathematical reasoning benchmarks (e.g., AIME, AMC); it remains unclear whether the findings generalize to other reasoning domains or more diverse tasks.

    The theoretical justification (Theorem 4.1) relies on a simplified tabular softmax bandit setting, which may not fully reflect the complexity of modern LLM training dynamics.

**Questions:**

How does the proposed direction-based extrapolation perform in non-mathematical reasoning tasks?

The token-replacement experiment convincingly shows that \Delta\log p identifies critical tokens. However, does this intervention sometimes harm performance? Are there cases were replacing a base model token with the RLVR model's choice leads to a wrong answer, and what characterizes those tokens?

---

> ### Author Response · Authors · 2025-11-21
> **[1/2] Rebuttal**
>
> >**Overall**
>
> Thank you for your constructive comments. Below, we address your questions one by one in detail. We have also **included all discussions below in our revision** (with the changed part marked in red).
>
> ---
> >**W1: "The test-time extrapolation method requires access to both the base and RLVR models, which may limit its practicality in settings where only the fine-tuned model is available."**
> ---
>
> Thank you very much for your question. Our paper does not contain any test-time extrapolation methods, and we would like to clearly highlight the novelty of our approach:
>
> **Three-stage training pipeline.** In the standard GRPO pipeline, the model first generates rollouts and then performs a single update using the GRPO objective. However, this one-shot update makes early training highly sample-inefficient and sensitive to noisy rollouts, increasing both training cost and instability.
> In contrast, our three-stage pipeline (fast → reposition → slow) reaches the best GRPO performance **many steps earlier** than vanilla GRPO. This results in **a substantial reduction in rollout usage and overall training budget**, while preserving the original GRPO objective and rollout process.
>
> **A plug-in, higher-level optimization mechanism.** SFPO does not require substantial modifications to GRPO because it operates at a higher level of the optimization pipeline. Unlike GRPO variants that introduce different twists to advantage computation, SFPO is **orthogonal** to these methods and can be seamlessly combined with them **without changing their training objective, KL regularization, or rollout procedure**. This makes SFPO an easy-to-use, plug-and-play mechanism applicable to any GRPO-family method.
>
> We have added additional details regarding novelty in Appendix D.1 in the revision.
>
> ---
> > **W2: "The paper focuses primarily on mathematical reasoning benchmarks (e.g., AIME, AMC); it remains unclear whether the findings generalize to other reasoning domains or more diverse tasks." "How does the proposed direction-based extrapolation perform in non-mathematical reasoning tasks?"**
> ---
> Thank you very much for raising this question regarding the generality of SFPO to other domains. We would like to humbly clarify that most reasoning RL works (e.g. GRPO, DAPO [1, 2] ) primarily focus on mathematical reasoning benchmarks, and we follow this established setting to ensure a fair and direct comparison with existing methods.
>
> However, we acknowledge that restricting the evaluation to math benchmarks does not fully demonstrate the generality of SFPO. Following your suggestion, We additionally trained DeepSeek-R1-Distill-Qwen-1.5B on the Skywork-OR1 Code RL training dataset [3] with 14.1k data. The training hyperparameters mirror those used for math tasks in submitted manuscript, with $K=3$ and $\alpha=0.8$ by default. We then evaluate the resulting checkpoints on LiveCodeBench [4] with a 32K response length and report Pass@1 (Avg 4). The results are summarized in the table below.
>
> |      | 0     | 50    | 100   | 150   | 200   | 250   | 300   | 350   | 400   |
> |------|-------|-------|-------|-------|-------|-------|-------|-------|-------|
> | GRPO | 17.02 |16.48 | 17.18 | 17.27 | 17.38 | 17.47 | 17.56 | 16.31 | 18.91 |
> | SFPO | 17.02 |**16.49**| **17.83** | **17.92** | **19.40** | **19.08** | **19.62** | **21.77** | **20.52** |
>
> As shown, SFPO consistently outperform GRPO throughout the whole training process. GRPO reaches a maximum Pass@1 (Avg 4) of 18.91 at 400 steps, whereas SFPO attains a higher peak performance of 21.77 at 350 steps. This consistent improvement on the coding benchmark provides additional evidence that the efficacy of SFPO extends beyond purely mathematical reasoning tasks.
>
> We also conduct ablation study to investigate the impact of $\alpha$ and $K$ on coding tasks. Similar to math tasks, we consider two scenarios: **(i)** a small inner-update budget $K = 3$ with varying $\alpha \in {0.2, 0.8, 1.0}$, and **(ii)** a larger budget $K = 7$ with the same set of $\alpha$ values. As shown in the table below, when $K=3$, the default value $\alpha=0.8$ achieves the best accuracy through out the training process, which is consistent with the findings in math tasks. While for $K=7$, a smaller $\alpha=0.2$ is preferred to mitigate stronger distribution drift and restore stability, again aligning with the trends observed in math RL training.
>
> | Setting                     |   0   |   50  |   100 |   150 |   200 |
> |----------------------------|------:|------:|------:|------:|------:|
> | GRPO                       | 17.02 | 16.48 | 17.18 | 17.27 | 17.38 |
> | SFPO (K = 3, α = 0.2)      | 17.02 | 17.47 | 17.47 | 18.10 | 18.28 |
> | SFPO (K = 3, α = 0.8)      | 17.02 | 16.49 | 17.83 | 17.92 | 19.40 |
> | SFPO (K = 3, α = 1.0)      | 17.02 | 17.56 | 18.54 | 17.20 | 16.95 |

---

> ### Author Response · Authors · 2025-11-21
> **[2/2] Rebuttal**
>
> | Setting                     |   0   |   50  |   100 |   150 |   200 |
> |----------------------------|------:|------:|------:|------:|------:|
> | GRPO                       | 17.02 | 16.48 | 17.18 | 17.27 | 17.38 |
> | SFPO (K = 7, α = 0.2)      | 17.02 | 18.01 | 19.44 | 22.49 | 22.13 |
> | SFPO (K = 7, α = 0.8)      | 17.02 | 17.02 | 19.53 | 20.43 | 19.35 |
> | SFPO (K = 7, α = 1.0)      | 17.02 | 16.75 | 16.67 | 18.01 | 18.27 |
>
> The above discussions and results have also been added to Appendix C.2 in the updated manuscript.
>
> ---
> >**W3: "The theoretical justification (Theorem 4.1) relies on a simplified tabular softmax bandit setting, which may not fully reflect the complexity of modern LLM training dynamics."**
> ---
> Thank you very much for your question. We would like to clarify that our paper does not contain a Theorem 4.1, and we are more than happy to discuss or expand on the theoretical intuition if you could provide additional details about the specific concern.
>
> ---
> >**W4: "The token-replacement experiment convincingly shows that \Delta\log p identifies critical tokens. However, does this intervention sometimes harm performance? Are there cases were replacing a base model token with the RLVR model's choice leads to a wrong answer, and what characterizes those tokens?"**
> ---
> Thank you very much for your question. Our paper does not contain any token-replacement experiment. We believe this comment may stem from a misunderstanding of our work, and we would like to clearly summarize our experiments in our paper:
>
> **Math RL training using SFPO:** We apply SFPO as a drop-in replacement for GRPO on five base LLMs, two math RL datasets with different scales, and six math reasoning benchmarks. SFPO consistently achieves improvements on benchamrks over baselines across different settings.
>
> **Training-dynamics analysis:** We track accuracy, reward, entropy, and response length over training and find that SFPO converges faster and exhibits more stable training behavior than GRPO.
>
> **Efficiency analysis:** We measure rollouts count, wall-clock time, and GPU memory, showing that SFPO reaches the same accuracy with substantially fewer rollouts and less time, while using similar memory as GRPO.
>
> **Ablation studies:** We vary the inner step count $K$, interpolation factor $\alpha$, Stage III, and the entropy-based schedule, confirming that small $K$ with moderately large $\alpha$ and slow correction are key to SFPO’s improvements.
>
> Additional experiments conducted during rebuttal:
>
> **Coding tasks:** SFPO also outperforms GRPO on coding RL benchmarks, confirming that its benefits extend beyond math reasoning.
>
> **Extension to DAPO:**  Applying SFPO on top of DAPO further improves both stability and final accuracy, even when zero-advantage issues are mitigated.
>
> **Compare Interpolation in Stage II with KL penalty:** Replacing Stage II interpolation with a KL penalty yields worse accuracy and less stable training, demonstrating the significance of our interpolation design in stage II.
>
> Since our method does not perform token-level replacement, the concern about harmful token interventions does not apply to SFPO.
>
> [1] Zhao, Z., et al. (2024). DeepSeekMath: Pushing the Limits of Mathematical Reasoning in Open Language Models. arXiv preprint arXiv:2402.03300.
>
> [2] Yu, Q., et al. (2025). DAPO: An Open-Source LLM Reinforcement Learning System at Scale. arXiv preprint arXiv:2503.14476.
>
> [3] He, J., et al. (2025). Skywork Open Reasoner 1 Technical Report. arXiv preprint arXiv:2505.22312.
>
> [4] Jain, N., et al. (2024). LiveCodeBench: Holistic and Contamination Free Evaluation of Large Language Models for Code. arXiv preprint arXiv:2403.07974.

---

### Official Review · Reviewer_UnHR · 2025-11-03

**Soundness:** 2
**Presentation:** 2
**Contribution:** 2
**Rating:** 2
**Confidence:** 4

**Summary:**

The paper proposes Slow–Fast Policy Optimization (SFPO), it's a three-stage modification to on-policy GRPO-style training for reasoning LLMs: (i) a fast trajectory of $K$ inner gradient steps on the same rollout batch, (ii) a reposition interpolation back toward the start point controlled by $\alpha$, and (iii) a slow correction extra step. See Alg. 1 (p. 3). Authors also introduce an entropy-triggered schedule that sets $\alpha \leftarrow 0$ after a $z$-score threshold on policy entropy (Eq. 11), which effectively reverts to GRPO near convergence.

**Strengths:**

- Simple, drop-in recipe that practitioners can try with minimal code churn (Alg. 1).
- Large rollout/time reductions to reach GRPO’s best accuracy (Fig. 4), aligning with the observation that rollouts dominate wall-clock.

**Weaknesses:**

1. GRPO-K: $K$ inner updates on the same batch without reposition or slow correction (a direct control for “just do more steps per batch”).

2. Lookahead-GRPO applies only the Stage II interpolation around GRPO.

3. Extra-grad-GRPO applies only the Stage III predictor–corrector step.

4. Clarify whether per-iteration compute (backprop) increases due to $K$; if so, compare at matched FLOPs (or matched wall-clock) and at fixed accuracy targets, not “GRPO’s best accuracy”.

5. The theoretical section is insufficient. Make Eq. (10) precise (define $c$ and $F(K,\alpha)$; assumptions on clipping/KL; conditions handling negative curvature), or clearly mark it as heuristic.

**Questions:**

See weaknesses.

---

> ### Author Response · Authors · 2025-11-21
> **[1/2] Rebuttal**
>
> >**Overall**
>
> We sincerely thank the reviewer for the time and effort spent reviewing our paper. Regarding the listed weaknesses, we would like to clarify that **Weaknesses 1–3 mainly restate aspects of our method and experimental setup, but do not identify concrete flaws or limitations of the approach itself**. For **Weaknesses 4 and 5**, the corresponding points are **already discussed in the main paper**, and we have further clarified them and added additional results in the revision as suggested.
>
> In light of these clarifications, we would be very grateful if you could kindly reconsider your overall assessment. We are also happy to address any additional concerns or to incorporate further related literature that you may recommend.
>
> ---
> >**W1-W3: "GRPO-K: K inner updates on the same batch without reposition or slow correction (a direct control for “just do more steps per batch”)." "Lookahead-GRPO applies only the Stage II interpolation around GRPO." "Extra-grad-GRPO applies only the Stage III predictor–corrector step."**
>
> ---
>
> We are sorry for the confusion. Hoewever, due to the weakness 1-3 are not complete statments we do not know your concern. We would happy to add any interpretations if you can provide more concrete concerns.
>
> ---
>
> >**W4: "Clarify whether per-iteration compute (backprop) increases due to K; if so, compare at matched FLOPs (or matched wall-clock) and at fixed accuracy targets, not “GRPO’s best accuracy”."**
>
> ---
>
> Thank you very much for this valuable concern. We clarify the computation cost and provide the requested comparisons at matched wall-clock time and fixed accuracy targets.
>
> **Does per-iteration compute increase with $K$?**
>
> Yes. In SFPO, each iteration performs $K$ fast policy updates on the same batch, so the backward-pass compute for the policy update scales approximately linearly with $K$. Previous works have shown that under GRPO, about **70%** of the per-step time is spent on rollout generation, **≈20%** on the policy update, and **≈10%** on other overhead. Increasing $K$ from 1 to 3 multiplies only the 20% “policy-update” portion, so the worst-case theoretical overhead is
>
> $$
> \frac{T_{\text{SFPO}}}{T_{\text{GRPO}}}
> \approx 1 + (K-1)\cdot 0.2
> = 1.4.
> $$
>
> In practice, measured average per-step wall-clock times on DeepSeek-R1-Distill-Qwen-1.5B are $\sim300$ s ($\sim240$ s for rollout generation, $\sim40$ s for actor update) for GRPO and $\sim410$ s ($\sim240$ s for rollout generation, $\sim150$ s for actor update) for SFPO (with $K=3,\alpha=0.8$), i.e., SFPO steps are only **≈1.21×** more expensive than GRPO steps. Moreover, our ablations on $\alpha$ and $K$ over both math and coding benchmarks show that this relatively small $K=3$ with a moderately large $\alpha=0.8$ is already sufficient; larger $K$ does **not** yield further gains and is therefore unnecessary in practice.
>
>
> **Matched wall-clock: fixed time budget.**
>
> To compare at matched compute, we convert training steps to elapsed wall-clock time and plot validation accuracy versus time as shown in in the Table below. On  DeepSeek-R1-Distill-Qwen-1.5B, for any reasonable time budget between 5 and 33 hours, SFPO consistently outperforms GRPO:
>
> | Time (h) | GRPO acc | SFPO acc | acc gain |
> |---------:|:--------:|:--------:|:-----------:|
> | 5        | 43.75   | 46.23   | **+2.48** |
> | 10       | 45.76   | 46.84   | **+1.08** |
> | 15       | 44.81   | 48.41   | **+3.60** |
> | 20       | 45.79   | 47.64   | **+1.85** |
> | 25       | 45.14   | 47.75   | **+2.61** |
> | 30       | 46.20   | 48.17   | **+1.97** |
> | 33       | 46.71   | 47.44   | **+0.74** |
>
> That is, **at the same wall-clock time** SFPO is typically **1–3.6 points** more accurate than GRPO, despite its relatively higher per-step compute.
>
> **Matched wall-clock: fixed accuracy targets.**
>
> We would first like to clarify that “GRPO’s best accuracy” in our original experiments was **already used as a fixed accuracy target**: we took the maximum validation accuracy achieved by GRPO during training and then measured how much wall-clock time SFPO needs to reach this same target.
>
> To make the comparison more flexible, we now also consider multiple fixed accuracy targets and ask how long each method needs to reach a given accuracy. On  DeepSeek-R1-Distill-Qwen-1.5B we obtain:
>
> | Target acc | $t_{\text{GRPO}}$ (h) | $t_{\text{SFPO}}$ (h) | Speedup (GRPO / SFPO) |
> |-----------:|:-----------------------:|:------------------------:|:----------------------:|
> | 0.44       | 5.78                    | 1.81                     | **3.19×**              |
> | 0.45       | 7.96                    | 2.21                     | **3.60×**              |
> | 0.46       | 29.55                   | 4.50                   | **6.57×**              |
> | 0.47       | 32.40                   | 10.56                  | **3.07×**              |
>
> Thus, to reach the same accuracy, SFPO requires **3–6.6× less wall-clock time**.

---

> ### Author Response · Authors · 2025-11-21
> **[2/2] Rebuttal**
>
> In summary, while Stage I of SFPO does incur additional gradient updates and thus increases per-iteration compute as $K$ grows, we deliberately operate in a regime with a small $K$ where the per-step overhead is modest. Under fair comparisons at matched wall-clock time and at fixed accuracy targets, SFPO is substantially more compute-efficient than GRPO, indicating that the extra Stage-I cost is more than compensated by improved sample and optimization efficiency.
>
> We have added the above analysis to the Appendix C.5 in the revision.
>
> ---
> >**W5: "The theoretical section is insufficient. Make Eq. (10) precise (define c and F(K, \alpha); assumptions on clipping/KL; conditions handling negative curvature), or clearly mark it as heuristic."**
>
> ---
>
> We are sorry for the confusion. Note that at the beginning of this subsection (lines 216–217), we explicitly introduce Eq. (10) as a **Theoretical intuition**, meaning that it is intended as intuition rather than a precise theoretical analysis.
>
> Following your suggestion, we have changed the heading from **Theoretical intuition** to **Intuition**, and we have updated the corresponding text in the revised main paper (lines 216–217) for clarity.

---

### Author Response · Authors · 2025-11-24
**Response to All Reviewers**

Dear reviewers,

We sincerely appreciate the time and effort you have devoted to providing thoughtful and constructive feedback. We are encouraged by your recognition of SFPO for its (1) "`well-structured and thoughtfully designed`" three-stage framework, which enables more stable, effective, and efficient policy optimization for reasoning-oriented RL training; (2) "`simplicity and ease of use`", in that SFPO is "`orthogonal`" to existing GRPO variants (e.g., DAPO) and can be seamlessly combined with them "`without modifying their training objective, KL regularization, or rollout procedure`"; and (3) demonstrated effectiveness and efficiency improvements, supported by a comprehensive set of experiments across a wide range of models and tasks.

In the sections below, we provide a brief summary of our responses to two common insightful questions from reviewers: the extension of SFPO to other GRPO variants and domains and the efficiency gains it provides.

---
> Common Concern A (reviwer JUEZ, 5eGS, GcTt) – The Generalization Ability of SFPO:
---
(1) *Extension to DAPO.*

When applied on top of DAPO, SFPO achieves higher accuracy across the entire training trajectory and reaches the target performance substantially earlier. This suggests that SFPO’s gains do not primarily stem from handling zero-advantage batches. Instead, SFPO operates at a higher-level optimization layer that is independent of the internal mechanics of GRPO or DAPO.

(2) *Extension to coding tasks.*

We also verify the efficacy of SFPO on coding tasks by demonstrating SFPO consistently outperforms GRPO throughout training and attains higher final accuracy on coding benchmarks. Additional ablations on coding data further show that the effects of the key hyperparameters $K$ and $\alpha$ closely match those observed in the mathematical domain, emphasizing the generalization ability and robust of SFPO across different domains.

---
> Common Concern B (reviwer GcTt, UnHR) – The Efficiency Gain of SFPO:
---
We acknowledge that the per-step compute of gradient updates increases as $K$ grows. However, in practice this overhead is modest because

- (1) the policy-gradient update accounts for only about 10–20\% of the wall-clock time per training step, and
- (2) a small value such as our default $K=3$ is already sufficient to demonstrate the effectiveness of SFPO across different models and tasks (both math and coding).

As a result, SFPO steps are only `≈1.21×` more expensive than GRPO steps, while SFPO reaches the same or higher accuracy substantially earlier in training process, leading to a significantly reduced total number of rollouts and wall-clock time.

In the original manuscript, we have already shown that, to reach a fixed accuracy (like the best accuracy achieved by GRPO), SFPO demonstrate substantially efficiency gain with `up to 4.93× fewer rollouts` and `4.19× less wall clock time`. In the rebuttal, we further shows that

- **At equal wall-clock time,** SFPO is typically `1–3.6 points more accurate` than GRPO, and
- **To reach the same accuracy (different levels)**, SFPO requires `3–6.6× less wall-clock time`.

We hope these clarifications fully address the reviewers’ concerns, and we will be more than happy to answer any of the reviewers' follow-up questions, if any.

Last but not least, we would also like to to sincerely thank the AC and SAC for their time and effort throughout the review process.

---

> ### Author Response · Authors · 2025-11-24
>
> ---
> > Appendix Updates:
> ---
> We append updated results and discussions to the appendix:
>
> C.1 shows that the interpolation-based reposition step in Stage II is crucial for SFPO.
>
> C.2 shows the consistent improvement of SFPO over GRPO on the coding tasks.
>
> C.3 shows the Stage II in SFPO provides better performance to a KL-constrained update.
>
> C.4 shows SFPO can further enhance the performance of DAPO.
>
> C.5 adds more discussions and results of training efficiency of SFPO.
>
> D.1 reemphasizes the novelty of SFPO.
>
> D.2 clarifies the naming rules of SFPO.
>
> D.3 provides more entropy analysis.
>
> D.5 provides the informal definition and analysis of $\mathcal{F}(F, \alpha)$ in Sec. 3.3.

---

### Meta-Review · Area_Chair_xVed · 2025-12-30

**Summary:**

Advantages:
1. The paper introduced a plug-and-play method Slow-Fast Policy Optimization (SFPO), a three-stage framework, including Fast Trajectory, Reposition, and Slow Correction, to ensure a more stable optimization process.
2. Empirical studies in math benchmarks showed the better performance of SFPO compared with GRPO.

Disadvantages:
1. Most reviewers raised the issues on the generalization of the proposed approach, which should be verified by experiments in the original version. Besides, the related contribution on theoretical analysis was over claimed.
2. The influence of parameters on the performance, and the efficiency were not clearly justified.

**Reviewer Concerns:**

Thanks for the rebuttal and the added experimental results.
The concern of Reviewer 5eGS was addressed by raising the score.
It lacked discussion with Reviewer UnHR to clarify the concern.

**Reviewer Scores:**

It received review scores 2, 6, 4, 6 in the first round. After rebuttal, Reviewer 5eGS increased the score from 4 to 6. The average score will be increased to 5.
Checking the review score 2 by Reviewer UnHR, the first 3 weaknesses were too short to show the weakness clearly. It is a pity that it did not have a chance to further discuss on that.

---

### Decision · Program_Chairs · 2026-01-26

Accept (Poster)